# Financial constraints and corporate bankruptcy risks in China: The buffer role of cash holdings

Quang Thu Luu[1], Hieu Thi Thanh Nguyen[2], Trang Ngoc Doan Tran[3], Tung Thanh Ho [ID][4]*

1 Faculty of Finance, Ho Chi Minh University of Banking, Ho Chi Minh, Vietnam, 2 School of Finance and International Business, Saxion University of Applied Sciences, Netherlands, 3 Faculty of Finance and Banking, Ho Chi Minh City Open University, Ho Chi Minh City, Vietnam, 4 Faculty of Finance and Banking, Ton Duc Thang University, Ho Chi Minh City, Vietnam

* hothanhtung@tdtu.edu.vn

## Abstract

Corporate bankruptcy risk in China is increasingly driven by structural credit discrimination and a systemic financial mismatch. This study investigates the impact of cash holdings and financial constraints on corporate bankruptcy risk in China. We employ the Two-step system Generalized Method of Moments (GMM) to analyze an unbalanced panel of 32,081 annual observations from listed firms in China, spanning the period from 2010 to 2023. Our findings indicate that higher financial constraints increase bankruptcy risk, as a one-point rise in the SA index reduces the Z-score by 4.26 points, supporting Market Timing Theory. Conversely, cash holdings serve as a powerful protective buffer; a 1% increase in cash holdings raises the Z-score by 0.37 points, supporting the Precautionary Savings and Trade-off theories. Furthermore, our results highlight the buffer role of cash holdings for financially constrained firms, where higher cash reserves mitigate the adverse effects of financial constraints on bankruptcy risk. Our main findings remain robust after employing alternative bankruptcy risk proxies, firm size-based, and exchange subsamples. These findings provide valuable insights for financial managers and policymakers, highlighting the importance of effective liquidity management and credit accessibility in mitigating corporate distress in emerging markets.

## 1. Introduction

Corporate bankruptcy risk is a significant concern for enterprises, investors, and policymakers, particularly in rapidly emerging economies such as China, where start-up businesses experience rapid growth and market volatility. Predicting bankruptcy risk is crucial for effective risk management and maintaining economic stability. One widely used measure is the Z-score model, developed by Altman [1], which has demonstrated an accuracy rate of 94% in forecasting corporate bankruptcy within two subsequent years. Given the increasing number of corporate defaults in China,

**Data availability statement:** The datasets generated during and/or analyzed during the current study are available in the Figshare Repository, https://doi.org/10.6084/m9.figshare.30953012.

**Funding:** This study is supported by Ton Duc Thang University, Ho Chi Minh City Open University, and Ho Chi Minh University of Banking. None of the authors received specific funding numbers for this study.

**Competing interests:** The authors have declared that no competing interests exist.

particularly among state-owned enterprises and major real estate firms in recent years (e.g., Evergrande and Country Garden), understanding the determinants of bankruptcy risk is essential.

A key factor influencing corporate bankruptcy risk is financial constraints. Theoretical perspectives on financial constraints and default risk remain divided. According to Market Timing Theory, firms ideally seek financing during favorable conditions; however, financially constrained firms lack this flexibility, facing a double bind where capital is most expensive when it is needed most [2]. While recent studies suggest that these constraints disproportionately threaten the viability of smaller firms [3], Agency Theory offers a contrasting view, suggesting that constraints may actually discipline managers, forcing them to be more cautious and scientific in their fund management [4]. In China, this tension is exacerbated by a financial mismatch, where political mandates often supersede market efficiency, leaving non-state-owned enterprises (NSOEs) structurally vulnerable to credit exclusion [5]. While previous studies have relied on accounting-based indices, such as the KZ, WW, and Z_FC, which suffer from endogeneity and distortions in the Chinese market, our study follows Hadlock and Pierce [6] and Yao and Yang [7] in utilizing the SA Index. By employing exogenous variables (firm size and age), the SA Index effectively captures institutional discrimination, ensuring our measurement reflects persistent structural barriers rather than transient fluctuations in financial ratios.

Similarly, the role of cash holdings is characterized by a fundamental tension between the immediate benefits of liquidity and the risks of mismanagement. According to Trade-off Theory, firms maintain cash as a vital buffer against operational shocks [8]. However, Agency Theory warns that abundant liquidity can lead to value-destroying projects or rent-seeking [9]. In China, this relationship is even more complex: while cash can be a panacea during crises [10], it can also serve as a signal of precautionary hoarding for firms anticipating a total loss of bank access [11]. Recent corporate defaults, including those of state-owned enterprises (SOEs) in 2020 and significant real estate firms in 2023, have highlighted growing concerns over bankruptcy risk. Despite its significance, most existing studies on cash holdings and bankruptcy risk focus on developed markets or a limited subset of Chinese companies, such as those listed on the Shanghai Stock Exchange [12]. A comprehensive analysis of the broader Chinese market remains limited. To address this, this study encompasses all A-shares of companies listed on the Shanghai Stock Exchange (SHSE) and the Shenzhen Stock Exchange (SZSE) to develop a generalized relationship between cash on hand and the likelihood of insolvency. Furthermore, while previous research predominantly relies on binary default proxies that misclassify government-sustained zombie firms, our study utilizes continuous measures, such as the Altman [13] Z-score and the Zmijewski [14] ZM-score. This approach captures the subtle, protective role of cash as an existential survival shield that standard binary models obscure.

To assess the bankruptcy risk of listed companies in China, we use the Fixed Effect Model (FEM) and the Two-step system Generalized Method of Moments (GMM). We select the sampling period from 2010 to 2023 to minimize the adverse

impacts of the 2008 financial crisis on the findings. Our findings show that while financial constraints increase the risk of corporate bankruptcy, cash holdings significantly reduce the risk of default. Our finding indicates that a percentage increase in cash holdings increases the Z-score by 0.37 points. However, our findings suggest that a 1-point increase in SA reduces the Z-score by 4.26 points.

Additionally, our study explores the moderating role of cash in the relationship between financial constraints and firm stability. We find that cash holdings significantly weaken the adverse effect of financial constraints on financial health. Our findings align with Aleksanyan and Huiban [2], Faulkender and Wang [9], and Zhitao and Xiang [5], Market Timing, Trade-offs, and the Precautionary Savings Theories. Ultimately, the robustness of our primary findings is enhanced by using ZM-score, Z'-score, and Z"-score as alternative proxies for bankruptcy risk.

We also follow Fama and French [15] in dividing our sample into different firm size subsamples, as well as Feder-Sempach et al. [16] in categorizing the sample by the SHSE and SZSE stock exchanges. Our robustness analysis reveals a significant institutional divide. For Small and Medium-sized firms and private firms listed on the SZSE, cash serves as a vital survival shield; due to discriminatory lending practices, these firms must rely on internal liquidity as a form of self-insurance to prevent immediate default. Conversely, for large firms and state-linked entities on the SHSE, the protective role of cash is negligible. These larger entities often benefit from "soft budget constraints," using cash more for operational compliance than for fundamental survival.

Our research complements previous research in the following ways. Firstly, this study is one of the first to examine the effects of cash holdings on bankruptcy risk in China. Secondly, prior research, such as Zhang et al. [12], focuses exclusively on A-share firms listed on the SHSE. In this study, we encompass all firms listed as A-shares on the SHSE and SZSE stock exchanges to comprehensively generalize the correlation between financial constraints, cash holdings, and bankruptcy risk. Thirdly, we add the interaction term between financial constraints and cash holding to test the buffer role of cash holding in reducing the corporate default risk. Prior studies have documented a relatively weak relationship between financial constraints, cash holdings, and the probability of default. Our findings indicate that financially constrained firms with a higher cash holdings ratio have lower corporate bankruptcy risk, which differs from prior studies. Finally, we confirm the robustness of our findings by utilizing the SA index to mitigate endogeneity and by applying alternative measures of bankruptcy risk. Our subsample tests across firm sizes and exchanges demonstrate that while financial constraints universally escalate risk, Small and Medium-sized firms, as well as Shenzhen-listed firms, are the most vulnerable to immediate default.

The paper is organized as follows: Section 2 creates the hypothesis, and Section 3 includes the data sources and methodology. Section 4 describes the findings. Section 5 discusses the findings, and Section 6 contains the conclusions.

## 2. Literature review

### 2.1. Financial constraints and bankruptcy risk

Financial constraints, or limitations on a firm's ability to obtain external financing, play a central role in determining corporate survival. When a firm cannot raise sufficient capital to fund its operations or investments at a reasonable cost, it faces significant friction that threatens its long-term viability. Market Timing Theory suggests that firms attempt to issue securities when market conditions are favorable. However, financially constrained firms lack the flexibility to time their financing decisions optimally. When constraints bind, particularly during market downturns or volatility, these firms face a double bind: they cannot access external capital when they need it most, and if they do secure financing, it often comes at prohibitively high costs. This financing inflexibility directly increases bankruptcy risks because the firm cannot smooth out operational shocks or meet debt maturity pressures [2,5].

Recent empirical studies largely support a positive relationship, confirming that financial constraints significantly increase the probability of corporate failure. Karas and Režňáková [17] found that for Small and Medium-sized firms, financial

constraints are critical predictors of default because these firms lack the capital buffers necessary to withstand bankruptcy risk. Le et al. [3] expand on this size-centric view in Vietnam, finding that financially constrained firms exhibit significantly higher bankruptcy risk due to their inability to access external capital buffers. They further clarify that this vulnerability effect is universal but disproportionately severe for smaller firms that lack the collateral to bypass credit rationing. This vulnerability is not limited to specific industries. Boateng et al. [18] argue that regardless of the nature or cause of the constraint, whether tax-related or structural, a firm is prone to "complete collapse" when faced with severe financing restrictions because it cannot cover compliance costs or operational liabilities. Furthermore, financial constraints serve as a critical transmission mechanism that amplifies the adverse effects of environmental and external shocks. When firms lack financing flexibility, they are unable to invest in adaptation strategies to mitigate climate change exposure and policy uncertainty [19–21]. Adamolekun [19] documents that financial constraints exacerbate bankruptcy risk for firms with high carbon emissions, as these firms cannot afford the necessary green investments to remain viable in a changing regulatory landscape.

In contrast, Agency theory offers a contrasting perspective. Jensen [22] argues that financial constraints can reduce bankruptcy risk by disciplining managerial behavior. When external financing is scarce, managers face stricter budget constraints and are compelled to be more risk-averse, as the consequences of a bad investment could be fatal for the firm. Yao et al. [4] argue that higher financial constraints act as a tight budget, motivating managers to use funds more cautiously and adopt scientific management strategies. Because these managers cannot afford to waste money on reckless projects, the firm operates more efficiently. As a result, this enforced discipline improves performance and decreases the likelihood of default compared to unconstrained firms that might take unnecessary risks.

China's financial system is characterized by significant state ownership influence and financial mismatch, where credit allocation is frequently driven by political mandates rather than pure market efficiency [5]. This creates a structural disadvantage for Non-State-Owned Enterprises (NSOEs), which face severe borrowing hurdles compared to State-Owned Enterprises (SOEs). Zhitao and Xiang [5] confirm that this institutional mismatch is a primary driver of default for private firms. In this context, financial constraints are not merely a friction but a systemic vulnerability. Given this unique institutional setting, a critical methodological gap exists in prior research regarding the accurate measurement of financial constraints. While accounting-based indices such as KZ (Kaplan-Zingales), WW (Whited-Wu), or the Z_FC index are widely established in general corporate finance research, Yao and Yang [7] explicitly argue that these traditional indices suffer from severe endogeneity because they rely on financial ratios like leverage and cash flow, which are prone to distortion by earnings management and state intervention. In China's dual-structure economy, a private firm's low leverage often reflects involuntary exclusion from the banking system rather than a conservative financial policy; thus, the KZ, WW, and Z_FC indices risk misclassifying the most structurally vulnerable firms as unconstrained. To resolve this endogeneity bias, our study follows Hadlock and Pierce [6] and Yao and Yang [7] by adopting the SA Index. The superiority of the SA Index lies in its construction solely from largely exogenous variables (firm size and age), allowing it to bypass the accounting noise inherent in the KZ and WW measures and directly proxy the structural discrimination described by Zhitao and Xiang [5], where state-owned banks systematically favor large, established entities. By utilizing the SA Index, this study ensures that the measurement of financial constraints captures the persistent, institutional barriers to survival rather than transient fluctuations in financial ratios. Our hypothesis is as follows:

H1: Financial constraints negatively correlate with the Z-score, implying that financially constrained firms have a higher bankruptcy risk in China.

## 2.2. Cash holdings and bankruptcy risk

The relationship between corporate cash holdings and bankruptcy risk is characterized by a fundamental tension between the immediate survival benefits of liquidity and the long-term systemic risks associated with capital mismanagement [9,23]. According to Trade-off Theory, firms determine an optimal cash level by balancing the marginal benefit of reduced financial distress against the opportunity costs of holding liquid assets, such as taxation on interest and potential agency

costs [9,23,24]. A central mechanism in this policy is the precautionary savings hypothesis, which posits that firms maintain cash as a vital buffer against adverse cash flow shocks, particularly when external financing is costly or inaccessible due to market frictions [8,10]. Furthermore, Pecking Order Theory suggests that information asymmetries lead firms to prioritize internal liquidity over debt or equity issuance to preserve financial flexibility and avoid financing traps where credit is withdrawn during periods of peak default risk [8,10,23].

However, Agency Theory warns that abundant liquid resources can exacerbate conflicts between managers and shareholders, providing incentives for managers to extract rents or invest in value-destroying projects [8,9]. Nguyen et al. [8] argue that during severe crises, the survival motive eclipses these concerns, prioritizing business resilience over efficiency. In such contexts, pre-crisis cash reserves become a primary determinant of business resilience, enabling firms to weather rapid revenue declines and maintain investment stability [8, 10, 25].

Analytically, this buffer mechanism operates through two distinct channels: short-term liquidity risk and long-term solvency risk. Cash holdings mitigate the former by providing a mechanical guarantee that the firm can meet its immediate debt obligations and payroll, even if revenue streams are temporarily interrupted [26]. Simultaneously, they address the latter by providing the strategic time necessary to restructure operations without the immediate threat of liquidation. By synthesizing these perspectives, it becomes clear that cash is not merely an idle asset but a critical tool for reducing a firm's distance to default and improving its overall financial health.

A significant tension between the buffer effect and the signaling effect characterizes the empirical relationship between cash and bankruptcy risk. Supporting the buffer perspective, Zheng [10] describes cash reserves as a panacea during the COVID-19 pandemic, finding that firms with high pre-crisis liquidity significantly outperformed their peers by avoiding the need to cut vital investments. This finding aligns with Nguyen et al. [8], who argue that internal liquidity is a primary driver of business resilience across 97 countries. From this perspective, cash is a direct tool for survival, reducing the probability of failure by ensuring that sudden revenue stops do not lead to immediate insolvency. In the European context, Yousaf and Briš [27] further show that higher liquidity ratios are consistently associated with better financial health and a lower likelihood of distress.

However, Poliakov and Zayukov [26] find that excessive liquidity can sometimes correlate with higher unprofitability, suggesting that holding too much idle cash might lead to operational inefficiencies that hurt a firm in the long run. Zhang et al. [11] found that for many A-share firms in the Chinese market, a rapid increase in cash holdings actually predicted an impending default. This suggests a precautionary hoarding behavior: firms that know they are in trouble might desperately pile up cash because they anticipate being cut off from bank loans. These findings refine the foundational view of Faulkender and Wang [9], who note that the marginal value of cash changes depending on the amount of debt a firm already carries.

Despite the established link between liquidity and survival, current literature is limited by a geographic bias toward developed economies and a methodological reliance on binary default proxies that are ill-suited for the Chinese institutional setting. Even recent studies examining bankruptcy in transition economies, such as Yousaf and Briš [27] in the Visegrad region, predominantly utilize binary logistic regression models where the dependent variable is a dichotomous default indicator. However, applying this binary framework to China is problematic due to the unreliability of standard distress proxies. Zhang et al. [11] demonstrate that the Special Treatment designation, often used as a binary proxy for default in A-share studies, primarily reflects accounting profitability rules rather than a terminal inability to meet debt obligations. Relying on such measures fails to capture the continuous erosion of financial health in a system where implicit government guarantees can sustain technically insolvent zombie firms. To bridge this gap, our study moves beyond binary classification by adopting the multivariate Z-score methodologies advocated by Altman [13] for non-manufacturing firms and emerging markets, alongside the bias-corrected ZM-score established by [14]. This approach provides a granular, continuous assessment of distance to default that captures the subtle protective role of cash buffers, which standard binary models obscure. Our hypothesis is as follows:

H2: Cash holdings positively affect the Z-score, implying lower corporate bankruptcy risk in China.

## 2.3. Financial constraints and cash holdings

The relationship between financial constraints and bankruptcy risk is not mechanical; it is contingent upon the availability of internal liquid resources to bridge periods of market exclusion. While financial constraints represent a blockade to external capital, cash holdings represent the immediate availability of internal capital. Integrating the Precautionary Savings motive with the Market Timing framework, we posit that cash holdings function as a critical substitution mechanism that attenuates the adverse impact of financial constraints on corporate survival. Theoretical literature suggests that the marginal value of cash is strictly decreasing with the accessibility of external finance [9]. For unconstrained firms, cash is merely one of several liquidity options; however, for financially constrained firms, it is often the sole means of survival against default. Denis and Sibilkov [28] argue that because constrained firms face prohibitively high external financing costs, they must rely on accumulated cash reserves to fund essential investments and service debt. In this context, cash holdings cease to be idle assets and become a strategic buffer that allows the firm to bypass the frictions of capital markets.

This buffering effect is particularly salient in the Chinese institutional setting, characterized by the financial mismatch [5]. In developed Western markets, firms may often rely on committed lines of credit as a backup liquidity source. However, in China's dual-structure economy, private firms usually face discriminatory lending policies that exclude them from such banking privileges. Consequently, when a Chinese private firm is financially constrained, it cannot rely on bank guarantees; instead, it must depend on self-insurance through cash holdings. Nguyen et al. [8] support this view, finding that during periods of systemic stress, internal liquidity is the primary determinant of resilience for firms that lack political connections or collateral.

Analytically, cash holdings moderate the relationship between constraints and bankruptcy by severing the transmission mechanism of distress. For a constrained firm with low cash reserves, the financial constraint binds tightly; any operational shock or debt maturity creates an immediate liquidity crisis that the firm cannot solve through borrowing (due to constraints) or internal payment (due to liquidity shortages), leading to a rapid erosion of the Z-score and high default risk [11]. Conversely, substantial cash holdings effectively decouple the firm's immediate survival from its ability to borrow externally. Even if the firm is structurally constrained and locked out of the credit market, high cash reserves allow it to smooth out cash flow volatility and meet obligations without triggering insolvency protocols [2]. Therefore, while financial constraints generally increase bankruptcy risk, this effect is non-linear. We posit that high cash holdings dampen the destructive power of financial constraints, serving as a protective shield that preserves financial health even when external capital channels are blocked. Based on this synthesis, we propose the following hypothesis:

H3: Financially constrained firms with higher cash holdings would decrease bankruptcy risk in China.

## 3. Data and methodology

### 3.1. Data

We use data from 2010 to 2023 in China to predict bankruptcy risk. We select the sampling period from 2010 to minimize the adverse impacts of the 2008 financial crisis on the findings. We collect data from the Taiwan Economic Journal (TEJ) database, a reliable data source in Taiwan. The sampling procedure followed previous studies. First, our sample includes the A-shares of listed firms in the SHSE and SZSE stock exchanges. Second, we follow Duong et al. [29] to exclude companies with missing accounting and financial data. We also follow Yu et al. [30] in trimming all variables at the 1% and 99% levels to minimize bias from extreme values. The final sample is an unbalanced panel consisting of 32,081 firm-year observations.

### 3.2. Variable definitions

Altman [1] recommends measuring a corporation's financial health using the Z-score. It is also an effective predictor for anticipating the default risk within two subsequent years. Accordingly, we adopt the Z-score as our primary measure of

bankruptcy risk. Additionally, we follow Zmijewski [14] and Altman [13] to calculate alternative bankruptcy risk, including ZM-score, Z'-score, and Z"-score, to examine the robustness of results.

The independent variables are the cash holdings ratio and financial constraint. Cash holdings refer to the cash and short-term investments a firm holds for its operational purposes. We follow Duchin [31] in calculating cash holdings as the total of cash and short-term investments divided by total book assets. In addition, we follow Hadlock and Pierce [6] and Yao and Yang [7] in constructing the SA index for financial constraints. The SA index (Size–Age index) is constructed to capture firms' financial constraints based on firm size and firm age, two relatively stable characteristics that are less volatile than traditional financial indicators. The equations are as follows:

$$SA\ index = (-0.737 \times Size) + (0.043 \times Size^2) - (0.040 \times Age)$$

Size is the natural logarithm of total assets, and Age is the difference between the observation year and the year of firm registration. A higher SA index value, particularly when approaching zero or becoming positive, reflects more severe financial constraints. Prior studies, including Hadlock and Pierce [6] and Yao and Yang [7], highlight the advantages of the SA index, noting that it is constructed from exogenous firm characteristics and is therefore less susceptible to endogeneity problems that frequently affect leverage- or cash flow-based measures such as the KZ, WW, and Z_FC index. The variable definitions are described in Appendix A.

### 3.3. Model construction

In this section, we follow the work of Le et al. [3] and Duong et al. [32,33] in investigating the impact of cash holdings and financial constraints on corporate default risk. In Model 1, we examine the effects of financial constraints, as measured by the SA index, and control variables on the risk of corporate bankruptcy.

$$Z - SCORE_{i.t} = a_0 + \beta_1 SA_{i,t} + \beta_2 CONTROL_{i,t} + a_i + a_t + \varepsilon_{i,t} \tag{1}$$

In Model 2, we follow Duchin [31] to examine the impact of cash holdings, as measured by CASH, and control variables on corporate bankruptcy risk.

$$Z - SCORE_{i.t} = a_0 + \beta_1 CASH_{i,t} + \beta_2 CONTROL_{i,t} + \delta_j + a_t + \varepsilon_{i,t} \tag{2}$$

In Model 3, we add the cash holding variable to examine the impact of financial constraints and cash holdings on corporate default risk.

$$Z - SCORE_{i.t} = a_0 + \beta_1 SA_{i,t} + \beta_2 CASH_{i,t} + \beta_3 CONTROL_{i,t} + \delta_j + a_t + \varepsilon_{i,t} \tag{3}$$

In Model 4, we add the interaction term (SA*CASH) to test whether cash holdings help mitigate default risk in financially constrained firms.

$$Z - SCORE_{i.t} = a_0 + \beta_1 SA_{i,t} + \beta_2 CASH_{i,t} + \beta_3 SA_{i,t} * CASH_{i,t} + \beta_4 CONTROL_{i,t} + \delta_j + a_t + \varepsilon_{i,t} \tag{4}$$

Besides, we replace the Z-SCORE with ZM-SCORE, Z'-SCORE, and Z"-SCORE to test whether our findings are robust after employing an alternative bankruptcy risk.

Where Z-SCORE is the bankruptcy risk coefficient of companies, the SA index is an indicator of financial constraints, and CASH is the cash holdings ratio of firms. In addition, the CONTROL variables include FAT, FTA, NIG, NPM, TAG, and ROA. The notation "t" represents time; the sign "i" denotes cross-sections; "α" denotes intercept, "j" denotes industry, and ε is the error term. Appendix A contains descriptions of every variable in the model.

## 3.4. Estimation method

To determine the appropriate estimation method (OLS, FEM, or REM) for each model, we conduct the Hausman test and the Redundant test. However, standard estimations like OLS, FEM, and REM may violate heteroskedasticity, autocorrelation, and endogeneity assumptions, which can substantially influence the findings [3,34]. To detect these issues, we use the modified Wald test for heteroskedasticity, the Wooldridge test for autocorrelation, and the Durbin–Wu–Hausman test for endogeneity. If violations are detected, the Two-step system Generalized Method of Moments is applied to address them. Additionally, we employ this method because corporate bankruptcy risk typically exhibits persistence; a firm's current bankruptcy risk is partly determined by its past financial health, due to the accumulation of debt and reputation effects. To ensure the validity of statistical inference, we use robust standard errors clustered at the firm level, applying the Windmeijer [35] finite-sample correction, which is essential for correcting the downward bias of standard errors in finite samples. Additionally, to prevent instrument proliferation that could weaken the Hansen test, we collapse the instrument set and limit the lag depth of the endogenous variables.

## 4. Results

### 4.1. Descriptive statistics

Table 1 reports descriptive statistics of the variables. This table includes the mean, max, min, standard deviation, and total observations. Table 1 gives an overview of statistics in the Chinese market. Table 1 indicates that the average value of the Z-score is about 5.11, with a standard deviation of 5.06. Altman [1] suggests that the Z-score is divided into three levels. The company has a serious distress risk if the Z-score is less than 1.81. If the Z-score is between 1.81 and 2.99, the firm is considered to have a low bankruptcy risk over the subsequent two years. If the Z-score exceeds 2.99, the business is in good financial health. As a result, the listed Chinese enterprises are generally in good financial condition. Compared to Le et al. [3] and Duong et al. [33], our sample shows stronger financial stability with average Z-scores of 2.53 and 2.46, respectively. The alternative bankruptcy proxies corroborate this finding of financial resilience. The ZM-score reports a mean of −2.21, which is notably lower than the benchmark mean of −1.58 observed in Zmijewski [14], indicating a lower-than-average risk profile and strong stability. Likewise, the Z'-score (8.98) and Z"-score (12.23) far exceed their respective safety cutoffs of 2.90 and 2.60 Altman [13], confirming robust financial strength across the sample. Regarding financial constraints, our sample shows a slightly less negative mean value of −2.09, compared to the −2.69 reported by

**Table 1. Descriptive statistics.**

| Variables | Mean | Median | Max | Min | Std. Dev. | N |
|---|---|---|---|---|---|---|
| Z-SCORE | 5.112 | 3.459 | 40.045 | −0.904 | 5.057 | 32,081 |
| ZM-SCORE | −2.212 | −2.292 | 1.944 | −4.433 | 1.193 | 32,081 |
| Z'-SCORE | 8.980 | 8.474 | 29.268 | −3.523 | 4.451 | 32,081 |
| Z"-SCORE | 12.230 | 11.724 | 32.518 | −0.273 | 4.451 | 32,081 |
| SA | −2.089 | −2.169 | 0.782 | −4.487 | 0.746 | 32,081 |
| CASH | 0.215 | 0.177 | 0.710 | 0.016 | 0.141 | 32,081 |
| FAT | 6.749 | 3.407 | 159.854 | 0.355 | 12.333 | 32,081 |
| FTA | 0.202 | 0.179 | 0.640 | 0.003 | 0.133 | 32,081 |
| NIG | −0.251 | 0.026 | 10.194 | −22.649 | 2.231 | 32,081 |
| NPM | 0.068 | 0.068 | 0.481 | −1.246 | 0.135 | 32,081 |
| TAG | 0.188 | 0.099 | 2.454 | −0.350 | 0.322 | 32,081 |
| ROA | 0.044 | 0.042 | 0.232 | −0.287 | 0.057 | 32,081 |

Notes: Table 1 presents the descriptive statistics for the variables employed in this study. All variable definitions are reported in Appendix A.

Yao and Yang [7]. Our sample reports an average cash holding of 0.22 and a standard deviation of 0.14, respectively. In addition, Table 1 also describes the mean of FAT, FTA, NIG, NPM, TAG, and ROA as 6.75, 0.20, −0.25, 0.07, 0.19, and 0.04, respectively.

## 4.2 Pearson Correlation Matrix

Table 2 reports the Pearson correlation matrix between the variables. Most independent variables exhibit weak correlations, except for NPM and ROA, which show a relatively high correlation coefficient of 0.82. Therefore, we conduct the Variance Inflation Factor (VIF) test to check the multicollinearity problem. The mean VIF is less than 5, implying no multicollinearity issue in our study [3,36].

## 4.3. Regression results

Table 3 reports the estimated results of the bankruptcy determination of enterprises in China. Table 3 reports that the P-values of the Hausman test and the Redundant Fixed Effects Test are less than 0.001, so the Fixed Effects Model (FEM) is more suitable than Pool OLS and REM.

Unfortunately, the modified Wald and Wooldridge tests confirm the presence of heteroskedasticity and autocorrelation. While these violations do not bias the coefficients, they invalidate statistical inference under standard FEM. Additionally, this study builds upon the work of Duong et al. [29] in implementing the Durbin–Wu–Hausman test, as the model contains endogenous variables, which would otherwise result in biased estimates.

After performing the Durbin–Wu–Hausman tests, Table 4 indicates that SA, CASH, FTA, NIG, NPM, TAG, and ROA are seven endogenous variables, as their residual coefficients are statistically significant. Therefore, we follow Duong et al. [29] and Vuong et al. [36] in re-estimating our findings using GMM estimations and reporting the results in Table 5. The insignificant Hansen and AR(2) results confirm the validity of our instruments and the absence of second-order serial correlation.

The analysis of marginal effects in Table 6 and Fig 1 reveals the critical economic role of cash as a structural buffer. At low cash levels (below 0.20), the economic penalty of financial constraints is severe, directly eroding solvency. However, as cash holdings enter the high cash zone (0.50–0.70), a striking economic reversal is observed, where financial constraints begin to contribute positively to stability. Managerially, this indicates that cash is not merely an idle asset but a strategic decoupling mechanism. For constrained firms, accumulating high cash reserves effectively neutralizes the toxicity of external financing frictions. It enables managers to adopt a prudent risk-management stance, transforming the discipline of financial constraints into a competitive advantage in solvency, thereby insulating the firm from market imperfections.

**Table 2. Pearson correlation matrix.**

|  | SA | CASH | FAT | FTA | NIG | NPM | TAG | ROA | VIF |
|---|---|---|---|---|---|---|---|---|---|
| **SA** | 1 |  |  |  |  |  |  |  | 1.036 |
| **CASH** | −0.145*** | 1 |  |  |  |  |  |  | 1.276 |
| **FAT** | 0.027*** | 0.121*** | 1 |  |  |  |  |  | 1.236 |
| **FTA** | 0.067*** | −0.350*** | −0.421*** | 1 |  |  |  |  | 1.381 |
| **NIG** | 0.049*** | 0.063*** | 0.022*** | −0.015*** | 1 |  |  |  | 1.204 |
| **NPM** | 0.046*** | 0.239*** | −0.016*** | −0.060*** | 0.363*** | 1 |  |  | 3.108 |
| **TAG** | 0.026*** | 0.224*** | 0.075*** | −0.148*** | 0.149*** | 0.264*** | 1 |  | 1.171 |
| **ROA** | 0.042*** | 0.274*** | 0.048*** | −0.071*** | 0.403*** | 0.820*** | 0.343*** | 1 | 3.420 |

*Notes: Table 2 reports the Pearson correlations among variables. All variable definitions are reported in Appendix A * **, *** indicating significance at 10%, 5%, and 1%, respectively.*

**Table 3. Regression results using the FEM estimations.**

| Variables | Model 1 | Model 2 | Model 3 | Model 4 |
|---|---|---|---|---|
| SA | −1.4244*** | | −1.3202*** | −1.660*** |
| | (<0.001) | | (<0.001) | (<0.001) |
| CASH | | 2.4002*** | 1.8670*** | −0.0344 |
| | | (<0.001) | (<0.001) | (0.961) |
| SA*CASH | | | | −0.8353*** |
| | | | | (0.005) |
| FAT | −0.0023 | −0.0036 | −0.0021 | −0.0021 |
| | (0.386) | (0.186) | (0.438) | (0.442) |
| FTA | −2.0183*** | −0.5289* | −1.3176*** | −1.3215*** |
| | (<0.001) | (0.087) | (<0.001) | (<0.001) |
| NIG | −0.0478*** | −0.0477*** | −0.0453*** | −0.0451*** |
| | (<0.001) | (<0.001) | (<0.001) | (<0.001) |
| NPM | −0.4985* | −1.1359*** | −0.6141** | −0.6220** |
| | (0.085) | (<0.001) | (0.034) | (0.031) |
| TAG | −0.4429*** | −0.6039*** | −0.5544*** | −0.5552*** |
| | (<0.001) | (<0.001) | (<0.001) | (<0.001) |
| ROA | 20.9919*** | 22.3612*** | 20.7699 | 20.7881 |
| | (<0.001) | (<0.001) | (<0.001) | (<0.001) |
| C | 1.7501*** | 3.9300*** | 1.4616*** | 1.8051*** |
| | (<0.001) | (<0.001) | (<0.001) | (<0.001) |
| R-squared | 0.6520 | 0.6494 | 0.6530 | 0.6531 |
| Adjusted R-squared | 0.6046 | 0.6017 | 0.6057 | 0.6058 |
| F- statistic | 13.7593 | 13.6041 | 13.8150 | 13.8169 |
| Prob (F-statistic) | <0.001 | <0.001 | <0.001 | <0.001 |
| Hausman Test (prob) | <0.001 | <0.001 | <0.001 | <0.001 |
| Redundant test (prob) | <0.001 | <0.001 | <0.001 | <0.001 |
| Modified Wald Test (prob) | <0.001 | <0.001 | <0.001 | <0.001 |
| Wooldridge test (prob) | <0.001 | <0.001 | <0.001 | <0.001 |
| N | 32,081 | 32,081 | 32,081 | 32,081 |

*Notes:* Table 3 *presents the regression results from FEM estimations. All variable definitions are reported in Appendix A * \*\*, \*\*\* indicating significance at 10%, 5%, and 1%, respectively. P-values are in parentheses.*

**Table 4. Durbin – Wu- Hausman Test.**

| Variables | Durbin (score) $\chi^2$ | Wu-Hausman F |
|---|---|---|
| SA | 96.883*** | 97.193*** |
| CASH | 84.174*** | 84.403*** |
| FAT | 2.624 | 2.623 |
| FTA | 12.747*** | 12.748*** |
| NIG | 16.076*** | 16.079*** |
| NPM | 18.002*** | 18.006*** |
| TAG | 51.742*** | 51.821*** |
| ROA | 81.291*** | 81.503*** |

*Notes:* Table 4 *presents the results of the endogeneity test. The findings indicate that the following variables are endogenous: SA, CASH, FTA, NIG, NPM, TAG, and ROA. The symbols \*\*\*, \*\*, and \*, respectively, are significant at 1%, 5%, and 10%.*

**Table 5. Regression results using the dynamic system GMM estimations with Z_SCORE.**

| Variables | Model 1 | Model 2 | Model 3 | Model 4 |
|---|---|---|---|---|
| Z-SCORE (−1) | 0.4296*** | 0.4635*** | 0.4295*** | 0.4652*** |
| | (<0.001) | (<0.001) | (<0.001) | (<0.001) |
| SA | −1.4683*** | | −1.3847*** | −4.2586*** |
| | (0.001) | | (<0.001) | (<0.001) |
| CASH | | 4.2371 | 0.3940 | 37.2763** |
| | | (0.346) | (0.825) | (0.015) |
| SA*CASH | | | | 16.3102** |
| | | | | (0.013) |
| FAT | 0.0005 | −0.0071* | 0.0014 | 0.0006 |
| | (0.918) | (0.190) | (0.766) | (0.914) |
| FTA | 0.6664 | 1.8841 | 1.1092 | 0.9412 |
| | (0.528) | (0.191) | (0.336) | (0.427) |
| NIG | 0.4095* | 0.5401** | 0.4467** | 0.3541* |
| | (0.056) | (0.021) | (0.031) | (0.091) |
| NPM | −6.2278* | −6.8141*** | −6.8222*** | −6.7673*** |
| | (0.062) | (0.009) | (0.010) | (0.010) |
| TAG | −1.2396 | 2.8524 | −1.5846 | −1.4197 |
| | (0.631) | (0.594) | (0.472) | (0.479) |
| ROA | 23.7506*** | 14.3890 | 24.9807*** | 23.5994*** |
| | (0.007) | (0.251) | (0.001) | (0.001) |
| Year Fixed | Yes | Yes | Yes | Yes |
| Industry Fixed | Yes | Yes | Yes | Yes |
| Prob AR (1) | <0.001 | <0.001 | <0.001 | <0.001 |
| Prob AR (2) | 0.992 | 0.573 | 0.996 | 0.874 |
| Hansen test | 0.005 | 0.003 | 0.007 | 0.125 |
| Instrument Rank | 41 | 41 | 43 | 45 |
| N | 26,957 | 26,957 | 26,957 | 26,957 |

*Notes:* Table 5 *presents the regression results from GMM estimations of the Z-score. All variable definitions are reported in Appendix A * \*\*, \*\*\* indicating significance at 10%, 5%, and 1%, respectively. P-values are in parentheses.*

## 4.4 Robustness test

To ensure that our findings are not driven by the specific construction of the Altman Z-score standard, we conduct a robustness test using three alternative proxies for corporate bankruptcy risk. The standard Z-score may not fully capture the nuances of the Chinese institutional setting or the "distance-to-default" for non-manufacturing firms. Therefore, following the methodologies of Altman [13] and Zmijewski [14], we re-estimate our baseline dynamic System GMM model using the ZM-score, Z'-score, and Z"-Score as dependent variables in Table 7.

Furthermore, following Fama and French [15], we examine whether the effects of financial constraints and cash holdings differ across firm sizes. Given that small and medium-sized firms often face greater barriers to external financing due to information asymmetry, limited collateral, and weaker ties with financial institutions, we test whether financial constraints and liquidity management strategies affect small and large firms differently. Table 8 presents the robustness test results for small, medium-sized, and large firms.

To further validate our findings within the unique institutional context of China, we conduct a sub-sample analysis by splitting the data between the SHSE and SZSE. The Chinese financial system is characterized by a financial mismatch

**Table 6. The average marginal effects of financial constraints (SA) on the Z-score at different levels of cash holdings (CASH).**

| Level of CASH | Marginal effect of SA (dy/dx) | Delta-method Std. Err | P-value | 95% conf. interval |
|---|---|---|---|---|
| 0.00 | −4.259*** | 1.180 | <0.001 | [-6.571; -1.946] |
| 0.10 | −2.628*** | 0.564 | <0.001 | [-3.733; -1.522] |
| 0.20 | −0.997*** | 0.339 | 0.003 | [-1.661; -0.332] |
| 0.30 | 0.635 | 0.885 | 0.473 | [-1.099; 2.368] |
| 0.40 | 2.266 | 1.524 | 0.137 | [-0.720; 5.252] |
| 0.50 | 3.897* | 2.175 | 0.073 | [-0.366; 8.159] |
| 0.60 | 5.528* | 2.830 | 0.051 | [-0.019; 11.075] |
| 0.70 | 7.159** | 3.487 | 0.040 | [0.324; 13.993] |

*Notes: The symbols \*\*\*, \*\*, and \*, respectively, are significant at 1%, 5%, and 10%.*

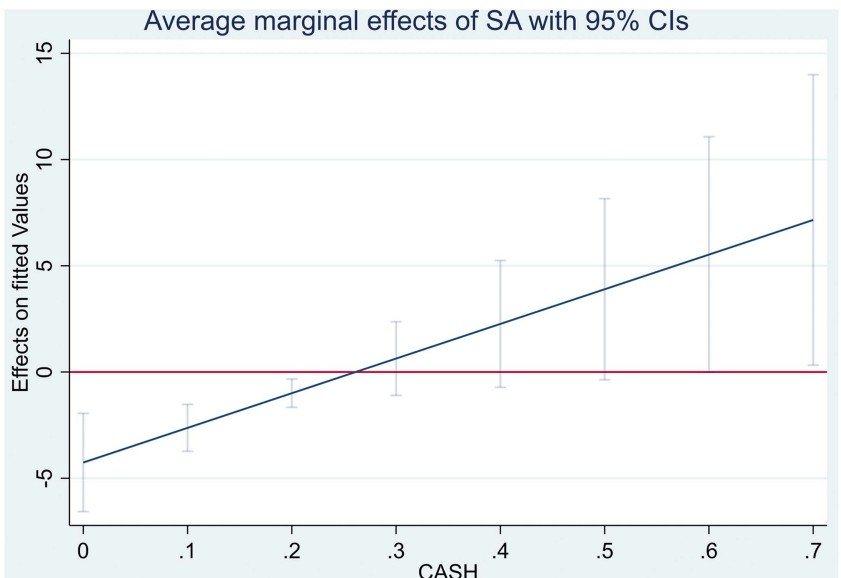

**Fig 1. The average marginal effects of financial constraints (SA) on the Z-score at different levels of cash holdings (CASH) with 95% CIs.**

where credit allocation is often driven by political mandates rather than market efficiency [5]. According to Feder-Sempach et al. [16], large and state-owned firms dominate the SHSE, while the SZSE features more private and innovation-driven firms. Zhitao and Xiang [5] state that these private entities face discriminatory lending policies and are structurally more vulnerable to default when access to external capital is restricted. Table 9 presents the robustness results across these two exchanges, testing whether the protective role of cash is more pronounced in environments with weaker institutional support.

## 5. Discussion

Table 5 reports that financial constraints increase the bankruptcy risk in Model 4. Specifically, one more point of SA reduces the Z-score by 4.26 points, implying higher bankruptcy risk. This finding aligns with the Market Timing Theory and the double bind mechanism described in our literature review. Constrained firms, lacking the flexibility to time their financing decisions, are forced to operate with suboptimal capital structures. When faced with market volatility or operational

**Table 7. Regression results using the dynamic system GMM estimations with ZM-SCORE, Z'-SCORE, Z"-SCORE.**

| Variables | ZM-SCORE | Z'-SCORE | Z"-SCORE |
|---|---|---|---|
| ZM-SCORE (−1) | 0.8343*** | | |
| | (<0.001) | | |
| Z'-SCORE (−1) | | 0.6587*** | |
| | | (<0.001) | |
| Z"-SCORE (−1) | | | 0.6618*** |
| | | | (<0.001) |
| SA | 0.6630*** | −2.1059** | −1.9729** |
| | (0.002) | (0.013) | (0.023) |
| CASH | −8.4054*** | 21.6677* | 19.7320* |
| | (0.005) | (0.054) | (0.088) |
| SA*CASH | −3.3784*** | 10.5386** | 9.7653** |
| | (0.008) | (0.028) | (0.047) |
| FAT | 0.0016 | 0.0424*** | 0.0423*** |
| | (0.218) | (<0.001) | (<0.001) |
| FTA | 0.1688 | 1.9913** | 1.9985** |
| | (0.547) | (0.040) | (0.038) |
| NIG | −0.2035*** | 0.5590*** | 0.5937*** |
| | (<0.001) | (<0.001) | (<0.001) |
| NPM | 1.5737** | −0.8654 | −0.9653 |
| | (0.049) | (0.671) | (0.638) |
| TAG | 0.7437** | −5.9725*** | −6.0841*** |
| | (0.020) | (<0.001) | (<0.001) |
| ROA | 0.5217 | 19.3971*** | 19.7418*** |
| | (0.726) | (<0.001) | (<0.001) |
| Year Fixed | Yes | Yes | Yes |
| Industry Fixed | Yes | Yes | Yes |
| Prob AR (1) | <0.001 | <0.001 | <0.001 |
| Prob AR (2) | 0.499 | 0.762 | 0.738 |
| Hansen test | 0.112 | 0.234 | 0.300 |
| Instrument Rank | 45 | 45 | 45 |
| N | 26,957 | 26,957 | 26,957 |

*Notes: Table 7 presents the robustness results from GMM estimations of ZM-score, Z'-score, and Z"-score. All variable definitions are reported in Appendix A * **, *** indicating significance at 10%, 5%, and 1%, respectively. P-values are in parentheses.*

shocks, these firms are unable to access external capital to smooth their cash flows, leading to a rapid deterioration in solvency [2,3]. Our results confirm that, in the Chinese context, the inability to borrow is not merely a friction but a primary driver of corporate failure for private and smaller entities. This result also supports Hypothesis H1.

Table 5 reports that the cash holdings ratio reduces the corporate default risk of listed firms in all models. Our findings indicate that a 1 percent increase in cash holding is predicted to result in a 0.37-point increase in the Z-score, implying a decrease in bankruptcy risk. This result supports Hypothesis H2 and is consistent with both the Trade-off theory and the Precautionary Savings Hypothesis. For Chinese firms, the benefits of holding cash as a safety buffer outweigh the costs. As Nguyen et al. [8] explain, this is likely because internal cash allows firms to survive when they cannot borrow money

**Table 8. Robustness test by employing different firm sizes.**

| Variables | Small and Medium-sized firms | | | | Large-sized firms | | | |
|---|---|---|---|---|---|---|---|---|
| | Z-SCORE | ZM-SCORE | Z'-SCORE | Z"-SCORE | Z-SCORE | ZM-SCORE | Z'-SCORE | Z"-SCORE |
| Z-SCORE (−1) | 0.4498*** | | | | 0.6200*** | | | |
| | (<0.001) | | | | (<0.001) | | | |
| ZM-SCORE (−1) | | 0.7001*** | | | | 0.8221*** | | |
| | | (<0.001) | | | | (<0.001) | | |
| Z'-SCORE (−1) | | | 0.5966*** | | | | 0.7988*** | |
| | | | (<0.001) | | | | (<0.001) | |
| Z"-SCORE (−1) | | | | 0.5991*** | | | | 0.8016 |
| | | | | (<0.001) | | | | (<0.001) |
| SA | −4.7166* | 1.6869** | −4.0876* | −4.0065* | −1.7320*** | 0.1850 | −0.3020 | −0.1792 |
| | (0.079) | (0.029) | (0.059) | (0.071) | (0.005) | (0.406) | (0.601) | (0.753) |
| CASH | 39.7060 | −21.0550** | 50.6175* | 48.9194* | 14.9770*** | −1.7407 | 0.5032 | −0.2858 |
| | (0.248) | (0.043) | (0.068) | (0.085) | (0.008) | (0.309) | (0.924) | (0.956) |
| SA*CASH | 17.2579 | −8.3906** | 21.5784* | 20.8580* | 7.8338** | −0.8911 | 0.2437 | −0.2605 |
| | (0.222) | (0.048) | (0.054) | (0.068) | (0.014) | (0.367) | (0.934) | (0.928) |
| FAT | −0.0016 | 0.0034* | 0.0370*** | 0.0372*** | 0.0101** | −0.0014 | 0.0442*** | 0.0435*** |
| | (0.819) | (0.070) | (<0.001) | (<0.001) | (0.015) | (0.352) | (<0.001) | (<0.001) |
| FTA | 0.6736 | 0.2360 | 1.6884 | 1.7945 | 1.6189* | −0.5891 | 4.4493*** | 4.3434*** |
| | (0.686) | (0.514) | (0.168) | (0.135) | (0.086) | (0.147) | (<0.001) | (<0.001) |
| NIG | 0.4931 | −0.2279*** | 0.5035** | 0.5097** | 0.0434 | −0.1528** | 0.2925** | 0.3042** |
| | (0.122) | (0.001) | (0.039) | (0.033) | (0.734) | (0.012) | (0.035) | (0.031) |
| NPM | −8.9436** | 1.1621 | −3.1812 | −2.8539 | −0.0226 | 0.9655 | −1.6614 | −1.9947 |
| | (0.026) | (0.285) | (0.302) | (0.340) | (0.990) | (0.339) | (0.426) | (0.355) |
| TAG | −6.1959** | 1.1017** | −7.9892*** | −7.7398*** | 1.7367 | 0.9565* | −0.2440 | −0.4717 |
| | (0.034) | (0.014) | (<0.001) | (<0.001) | (0.195) | (0.074) | (0.869) | (0.749) |
| ROA | 37.8401*** | −0.6684 | 27.0930*** | 26.0665*** | 1.1988 | −2.8720* | 9.7913** | 10.6651** |
| | (0.001) | (0.783) | (<0.001) | (<0.001) | (0.752) | (0.079) | (0.044) | (0.031) |
| Year Fixed | Yes | Yes | Yes | Yes | Yes | Yes | Yes | Yes |
| Industry Fixed | Yes | Yes | Yes | Yes | Yes | Yes | Yes | Yes |
| Prob AR (1) | <0.001 | <0.001 | <0.001 | <0.001 | <0.001 | <0.001 | <0.001 | <0.001 |
| Prob AR (2) | 0.806 | 0.304 | 0.670 | 0.713 | 0.756 | 0.819 | 0.210 | 0.230 |
| Hansen test | 0.059 | 0.154 | 0.935 | 0.863 | 0.869 | 0.206 | 0.127 | 0.119 |
| Instrument Rank | 45 | 45 | 45 | 45 | 45 | 45 | 45 | 45 |
| N | 18,131 | 18,131 | 18,131 | 18,131 | 7,855 | 7,855 | 7,855 | 7,855 |

*Notes:* Table 8 presents the robustness results from GMM estimations of different firm-sized samples. All variable definitions are reported in Appendix A * **, *** indicating significance at 10%, 5%, and 1%, respectively. P-values are in parentheses.

from banks. Poliakov and Zayukov [26] also support this view, noting that cash helps companies meet immediate debt obligations and gives them time to restructure during difficult times. Although Zhang et al. [11] argued that piling up cash could be a warning sign of distress, our results show that for the majority of listed firms, high cash reserves are a necessary protection that significantly improves financial stability.

Table 5 reports that the interaction term SA*CASH increases the Z-score, implying a buffer role of cash holding in reducing the corporate default risk of financially constrained firms. The partial effect of SA on Z-score, while holding other factors constant, is measured by the following function:

**Table 9. Robustness test by splitting the Shanghai and Shenzhen Stock Exchanges.**

| Variables | Shanghai | | | | Shenzhen | | | |
|---|---|---|---|---|---|---|---|---|
| | Z-SCORE | ZM-SCORE | Z'-SCORE | Z"-SCORE | Z-SCORE | ZM-SCORE | Z'-SCORE | Z"-SCORE |
| Z-SCORE (−1) | 0.4394*** | | | | 0.5009*** | | | |
| | (<0.001) | | | | (<0.001) | | | |
| ZM-SCORE (−1) | | 0.8723*** | | | | 0.7869*** | | |
| | | (<0.001) | | | | (<0.001) | | |
| Z'-SCORE (−1) | | | 0.6530*** | | | | 0.6704*** | |
| | | | (<0.001) | | | | (<0.001) | |
| Z"-SCORE (−1) | | | | 0.6517*** | | | | 0.6795*** |
| | | | | (<0.001) | | | | (<0.001) |
| SA | 0.4867 | 0.3470 | −1.5154* | −1.4892* | −5.0403*** | 0.5100* | −2.6381** | −2.4279* |
| | (0.619) | (0.119) | (0.073) | (0.080) | (0.005) | (0.065) | (0.038) | (0.068) |
| CASH | −19.0055* | −4.4715* | 8.5464 | 8.2981 | 46.6702** | −5.8924 | 30.0070* | 27.0379 |
| | (0.095) | (0.070) | (0.356) | (0.373) | (0.041) | (0.103) | (0.076) | (0.125) |
| SA*CASH | −9.5167* | −1.6476 | 4.2443 | 4.1504 | 20.9042** | −2.2185 | 14.3814** | 13.2398* |
| | (0.077) | (0.133) | (0.315) | (0.328) | (0.031) | (0.139) | (0.042) | (0.072) |
| FAT | −0.0016 | −0.0010 | 0.0569*** | 0.0566*** | −0.0002 | 0.0013 | 0.0343*** | 0.0342*** |
| | (0.828) | (0.625) | (<0.001) | (<0.001) | (0.976) | (0.233) | (<0.001) | (<0.001) |
| FTA | −0.9841 | −0.0018 | 1.3087 | 1.2530 | 2.0754 | 0.0490 | 2.4498** | 2.4611** |
| | (0.491) | (0.996) | (0.351) | (0.377) | (0.164) | (0.859) | (0.045) | (0.041) |
| NIG | 0.3442 | 0.0912 | 0.4545 | 0.4640 | 0.4709** | −0.1834*** | 0.4416*** | 0.4751*** |
| | (0.194) | (0.482) | (0.106) | (0.108) | (0.032) | (<0.001) | (0.009) | (0.006) |
| NPM | 1.3136 | 0.0217 | 3.8066 | 3.5350 | −8.0594*** | 0.8684 | −0.9390 | −0.9554 |
| | (0.697) | (0.981) | (0.212) | (0.243) | (0.009) | (0.283) | (0.709) | (0.701) |
| TAG | 0.5296 | −0.0508 | −0.4102 | −0.5654 | −4.9608** | 0.6255** | −7.8145*** | −7.8881*** |
| | (0.818) | (0.938) | (0.823) | (0.758) | (0.027) | (0.033) | (<0.001) | (<0.001) |
| ROA | 7.8125 | 1.5360 | 2.5007 | 3.2133 | 35.5978*** | 1.4061 | 25.4887*** | 25.5939*** |
| | (0.366) | (0.594) | (0.740) | (0.667) | (<0.001) | (0.328) | (<0.001) | (<0.001) |
| Year Fixed | Yes | Yes | Yes | Yes | Yes | Yes | Yes | Yes |
| Industry Fixed | Yes | Yes | Yes | Yes | Yes | Yes | Yes | Yes |
| Prob AR (1) | <0.001 | 0.007 | <0.001 | <0.001 | <0.001 | <0.001 | <0.001 | <0.001 |
| Prob AR (2) | 0.349 | 0.548 | 0.749 | 0.794 | 0.950 | 0.520 | 0.453 | 0.399 |
| Hansen test | 0.011 | 0.254 | 0.092 | 0.099 | 0.743 | 0.007 | 0.924 | 0.961 |
| Instrument Rank | 45 | 45 | 45 | 45 | 45 | 45 | 45 | 45 |
| N | 9,567 | 9,567 | 9,567 | 9,567 | 17,390 | 17,390 | 17,390 | 17,390 |

*Notes: Table 9 presents the robustness results from GMM estimations of different exchange samples. All variable definitions are reported in Appendix A * **, *** indicating significance at 10%, 5%, and 1%, respectively. P-values are in parentheses.*

$$\frac{\Delta Z-score}{\Delta SA} = -4.2586 + 16.3102 * CASH.$$

The coefficient of SA*CASH is greater than zero, implying that CASH weakens the negative relationship between SA and Z-score. Specifically, when CASH is at the value of 0.5, the effect of SA on Z-score is −4.2586 + 16.3102*0.5 = 3.897 points. The finding supports Hypothesis H3, reflecting the integration of the Precautionary Savings motive and the Market-Timing framework. These theories suggest that constrained firms strategically accumulate cash to bridge periods of

market exclusion, thereby mitigating the adverse impact of financial constraints on corporate survival. This result aligns with Faulkender and Wang [9] and Denis and Sibilkov [28], who argue that because restricted firms cannot access external capital markets, internal liquidity becomes their primary source of funding for essential operations. In the context of the Chinese market, this finding is particularly relevant to the financial mismatch described by Zhitao and Xiang [5], where private enterprises often face discriminatory lending practices. Consequently, these firms rely on accumulated cash as a form of self-insurance to bypass banking restrictions. Nguyen et al. [8] and Aleksanyan and Huiban [2] note that this financial flexibility enables constrained firms to smooth their cash flows and decouple their survival from the availability of bank loans, effectively mitigating the risk of default during periods of credit exclusion.

Table 5 reports that ROA exerts a strong positive effect on the Z-score, consistent with Altman [1], who shows that bankruptcy risk declines significantly as return on assets increases. This finding suggests that firms with more efficient asset utilization tend to exhibit stronger financial stability. Additionally, our results indicate that NIG significantly mitigates bankruptcy risk. The positive effect of NIG supports the argument of Musa et al. [37] that stable growth in net income serves as a liquidity buffer, enabling firms to accumulate internal cash flows and better withstand financial shocks. In contrast, the findings reveal that a higher NPM is associated with an increased risk of bankruptcy risk. A trade-off in the profitability structure can explain this result, as firms with higher NPM tend to exhibit lower asset turnover when return on assets (ROA) is held constant. Slower inventory turnover and reduced cash flow flexibility weaken liquidity and, in turn, increase firms' vulnerability to financial distress. This interpretation is consistent with the arguments presented by Soliman [38].

Table 7 presents the robustness test results using three alternative proxies for corporate bankruptcy risk: the ZM-score, Z'-score, and Z"-Score. The results provide strong validation for our baseline findings. Although the coefficient signs differ between models due to construction (higher ZM-score indicate higher risk, whereas higher Z-score and Z"-score indicate better health), the economic implications remain identical. Across all specifications, financial constraints significantly increase bankruptcy risk, while cash holdings consistently reduce it. Crucially, the interaction term SA*CASH remains significant in every model, proving that our main finding is reliable: cash acts as a vital survival shield for constrained firms, regardless of the bankruptcy risk measure employed.

Table 8 reveals a notable split in how firm size affects default, with Small and Medium-sized firms being most sensitive to immediate default, while large firms are more sensitive to overall financial health. For Small and Medium-sized firms, the significant results in the ZM-score, Z'-score, and Z"-score models confirm that financial constraints directly increase the probability of immediate default, reflecting the acute financial mismatch and discriminatory lending described by Zhitao and Xiang [5]. In this context, the significant interaction term indicates that internal liquidity is a vital necessity for smaller firms, serving as a form of self-insurance to circumvent banking restrictions. Conversely, for large firms, the significant results in the Z-score model suggest that while constraints erode general financial health, they do not trigger immediate default as easily. Consequently, Small and Medium-sized firms utilize cash reserves to prevent immediate bankruptcy, whereas large firms use them to preserve long-term financial health.

Table 9 reports the robustness of our findings across the SHSE and SZSE. As noted by Feder-Sempach et al. [16], the SHSE is dominated by SOEs, whereas the SZSE hosts a higher concentration of private and innovation-driven firms. Our results reveal that the Z-score model is highly significant and statistically valid in the Shenzhen sample, confirming that for credit-sensitive private firms, cash is a vital shield for overall financial health. In contrast, the Z-score is less appropriate for the Shanghai sample due to the failed Hansen test (p = 0.011), likely because the soft budget constraints of SOEs decouple their financial health from market-based liquidity. Moreover, the significant interaction terms in the Z'-score and Z"-score models for Shenzhen confirms that internal liquidity acts as a vital survival shield for private firms. Conversely, in Shanghai, the interaction effect is statistically insignificantly across all models, suggesting that SOEs do not rely on cash buffers to survive financial constraints. Their privileged access to state credit effectively renders their survival independent of internal liquidity, rendering the self-insurance mechanism unnecessary.

## 6. Conclusion

The study investigates the effect of cash holdings and financing constraints on the bankruptcy risk of firms in China. Our sample comprises the A-shares of listed firms on the Shanghai and Shenzhen stock exchanges from 2010 to 2023. We employ the Two-step system GMM estimation to address heteroskedasticity, autocorrelation, and potential endogeneity issues. Our findings demonstrate that the cash holding ratio reduces corporate bankruptcy risk in China by serving as a buffer against bankruptcy risk. Specifically, reducing investment spending and boosting cash reserve ratios prevent financially constrained businesses from going bankrupt. Conversely, firms facing higher financial constraints are more susceptible to bankruptcy, emphasizing the critical role of external financing accessibility in corporate stability. To enhance robustness, we employ alternative bankruptcy risk measures, including ZM-score, Z'-score, and Z''-score. These results confirm that our main findings remain robust. Additionally, our results remain consistent across firm-size and stock exchange subsamples, further validating the vital role of liquidity management in mitigating corporate distress.

Our study offers significant advancements in understanding the dynamics of liquidity and distress within emerging economies. Beyond confirming the negative relationship between cash and bankruptcy, we identify the conditional value of cash as a strategic buffer. We demonstrate that cash holdings actively decouple financial constraints from immediate distress, acting as a crucial survival mechanism rather than merely a liquid asset. This finding refines the precautionary savings theory by situating it within an institutional context. In markets characterized by high financing frictions and information asymmetry, cash is not simply an operational requirement but a strategic shield against institutional voids that prevents constraints from escalating into insolvency.

The robustness of our results across distinct market segments offers nuanced implications for stakeholders in China and similar emerging markets. For managers, particularly in Small and Medium-sized firms, this implies that liquidity management must take precedence over aggressive expansion in times of credit tightening. For policymakers, the findings highlight a structural inefficiency. Firms are forced to hoard cash due to severe financing frictions. Policies aimed at deepening credit markets and reducing information asymmetry would lessen the necessity for this defensive cash hoarding, potentially unlocking capital for innovation.

Additionally, the findings reveal a systemic reliance on internal liquidity for policymakers. In the Shenzhen market, dominated by innovation-driven Small and Medium-sized firms, the necessity to hoard cash for survival implies a high opportunity cost, potentially crowding out R&D investment. Policy reforms prioritize alleviating financing discrimination against private firms to unlock this defensive capital for innovation. Conversely, in the Shanghai market, dominated by large SOEs, the insignificant role of cash holdings suggests that internal liquidity is not the primary buffer against financial distress for these entities. This likely reflects the prevalence of soft budget constraints and implicit government guarantees, which shield state-backed firms from liquidity shocks regardless of their cash levels. Therefore, policy focus should shift towards structural market-oriented reforms to reduce moral hazard, ensuring that SOEs' survival depends on operational efficiency rather than preferential access to external credit.

Although our study extends the financial management literature in emerging markets, it has the following limitations. First, the research primarily focuses on China, so the findings may not be applicable to other markets due to differences in market microstructures and institutional settings. Second, the classification of firms into large enterprises and Small and Medium-sized firms is based on size breakpoints, which may not fully capture the institutional, industrial, and economic realities of Chinese firms and could therefore result in some degree of misclassification. Third, although GMM is a robust and versatile estimation technique, it has inherent limitations, including issues with computational efficiency, sensitivity to outliers and initialization, reliance on distributional assumptions, and challenges in performance under weak identification. Future research could extend our framework by conducting a cross-country comparative study, employing alternative size definitions such as government SME standards or industry-adjusted thresholds, and exploring how variations in financial market development and institutional quality influence the relationship between financial constraints, cash holdings, and corporate bankruptcy risk.

To address these limitations, future research should adopt a comparative institutional approach. Specifically, scholars could investigate whether the buffering role of cash remains as potent in market-based systems, such as the US or the UK, where external financing is more accessible, compared to China's bank-based system. Additionally, future studies could extend our framework by exploring how specific institutional reforms, such as interest rate liberalization or the development of digital finance, alter the intensity of the liquidity–distress dynamic. Finally, subsequent research may employ alternative size definitions, such as government SME standards or industry-adjusted thresholds, to validate the robustness of the relationships identified in this study.

## Supporting information

**S1 Appendix. Variable definitions [39–44].**
(DOCX)

## Author contributions

**Conceptualization:** Quang Thu Luu, Tung Thanh Ho.

**Investigation:** Hieu Thi Thanh Nguyen, Trang Ngoc Doan Tran.

**Methodology:** Hieu Thi Thanh Nguyen, Trang Ngoc Doan Tran.

**Software:** Hieu Thi Thanh Nguyen, Trang Ngoc Doan Tran.

**Supervision:** Tung Thanh Ho.

**Writing – original draft:** Quang Thu Luu, Hieu Thi Thanh Nguyen, Tung Thanh Ho.

**Writing – review & editing:** Quang Thu Luu, Tung Thanh Ho.

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
