## [Decision Letter · Decision Letter 0]

2 Oct 2024

Dear Dr. Ho,

Thank you for submitting your manuscript to PLOS ONE. After careful consideration, we feel that it has merit but does not fully meet PLOS ONE’s publication criteria as it currently stands. Therefore, we invite you to submit a revised version of the manuscript that addresses the points raised during the review process.

We look forward to receiving your revised manuscript.

Kind regards,

Vanessa Carels

Staff Editor

PLOS ONE

Journal Requirements:

“This study is supported by Ton Duc Thang University. None of the authors receive specific funding number for this study.”

Reviewers' comments:

Reviewer's Responses to Questions

**Comments to the Author**

1. Is the manuscript technically sound, and do the data support the conclusions?

Reviewer #1: Partly

Reviewer #2: Partly

2. Has the statistical analysis been performed appropriately and rigorously?

Reviewer #1: Yes

Reviewer #2: Yes

3. Have the authors made all data underlying the findings in their manuscript fully available?

Reviewer #1: No

Reviewer #2: Yes

4. Is the manuscript presented in an intelligible fashion and written in standard English?

Reviewer #1: Yes

Reviewer #2: Yes

Reviewer #1: The authors provide a worthy contribution to the dense discussion on the impact of cash holdings and financial constraints on firms' bankruptcy risks. Specifically, they focus on 25,002 firm-year observations in the period of 2010-2023, examining only publicly listed A-share companies on the Shanghai and Shenzen stock exchanges. The findings they present align with the expectations of both the market timing theory and the trade-off theory - large financially constrained firms with large cash holdings exhibit lower bankruptcy risk with their Z-score (the primary measure of risk in the paper) rising by ~2.2 points (going by the GMM estimations). The paper is well-structured, concise and tries to lend scrutiny to the authors' findings. However, certain parts of the paper could use improvement.

1. Zhang et al. [9] provides an analysis similar to yours, but only for the Shanghai stock exchange. Your results corroborate their findings, however I would also be interested if there are any differences in the effects you find across the Shanghai and Shenzen stock exchanges. Is the "listing location" something that is available to you? Ideally, you can include a firm's location proxy in the regressions as an additional fixed effect, or you can create heterogeneous effects based on these.

2. As a robustness measure you introduce the adjusted WW index along with the KZ index which are both proxies for financial constraint. However, you omit the AWW from table 2 and table 6. What is the reasoning behind this?

3. You note: "We also follow Le et al. [24] to winsorize all variables at the 5% and 95% levels to minimize extreme values bias". I consider a winsor that applies to 10% of the observations you use to be at the very least unrepresentative by itself, regardless of the paper you refer to, or the amount of the data at your disposal. Winsoring the data is of course permitted, but I would advise against going over the 2.5% border on either side. Also, you maintain a great deal of the observations you start with even with the usual lag observation losses in GMM - this allows you to trim your sample and avoid any potential confusion. I would wish to see:

• the regression results without any winsoring/trimming,

• the results when winsoring the data on the top and bottom 1% symmetrically, and

• the results when trimming the observations the same way.

Ideally, one of these should be the main paper estimates, or you should provide a more sufficient reason why you winsor on a total of 10% of observations.

4. You split the firms into the manufacturing and non-manufacturing industries, you use the ISG (industry sales growth) in your financial constraint index creation, and you use the industry fixed effects in the regressions. However, you do not note what industry variable you base this on? Is this always a 2-digit sector, as in Chan et al. [5] you reference, or is this something else? This should be made clear and consistent throughout the paper.

5. Another point regarding your split on the manufacturing and non-manufacturing groups of firms. It is true that your results align with the Altman's paper, however your observation count in table 6 is rather odd. 18,367 vs. 1,729 observations seems a quite unbalanced comparison. How do you actually split the firms into these two groups, and is there any way you could improve this split (in terms of observation count balance) with the data available to you?

6. My last major point is your conclusion. Throughout your paper you note that (based on the literature so far) SME's and large firms differ in the effects financial constraints and cash holdings have on the bankruptcy risk of said firms. I assume, since you have not provided any non-ratio descriptive statistics on the firms in your sample, that A-share listed companies are by and large non-SME's. However, you conclude with the practical implications that pertain exactly to SMEs. I would be wary of such interpretations without any "raw" descriptive statistics nor SME-specific heterogeneous effects on the firms in your sample. You should provide the readers with more context on the firms you work with, at the very least for all the starting variables you use in your analysis before any ratios.

Minor notes:

• You write the 3.4. section as if it was done in advance. In my opinion it makes it only harder to read and understand, with little benefit to it.

• You have two H2 hypotheses, I don't know if this is intentional or not, but I suggest you change it to H2 and H3.

• "We select the sampling period is from 2010 to prevents tha adverse impacts of the financial crisis in 2008 on the findings." - typo on pg. 3 and pg. 8.

Reviewer #2: The introduction section is poorly managed. It is likely a literature review. It requires re-writing: The introduction section of a typical research is, needless to say, like a driving force that leads to the rest of the paper and thus it should be carefully written. The introduction should properly present (in no particular order but in a logical manner) the background information, the motivation of the study, the research questions, the gaps in the literature that the study fills, contribution of the study and so on. It is suggested to re-write the introduction in order to make it sufficient, better sequenced and having read smoothly for the readers.

The authors have discussed two alternative theories supporting the relationship between cash holdings and bankruptcy risk (hypothesis 1). However, they propose a single relationship in the hypothesis. I think that it is better to propose two alternative hypotheses relating to each theory. The same case for hypothesis 2.

The author(s) need to link their findings more strongly to context, highlight their economic, academic/research and policy/practice implications

**Do you want your identity to be public for this peer review?** For information about this choice, including consent withdrawal, please see our Privacy Policy

Reviewer #1: No

Reviewer #2: **Yes:** Moncef Guizani

---

## [Author Response · Author response to Decision Letter 1]

20 Feb 2025

RESPONSES TO REVIEWER'S REPORT

PONE-D-24-39978

FINANCIAL CONSTRAINTS AND CORPORATE BANKRUPTCY RISKS IN CHINA: THE BUFFER ROLE OF CASH HOLDINGS

PLOS ONE

Dear editorial office and anonymous reviewers,

Thank you for allowing us to revise and resubmit our paper to your PLOS ONE. Thank you for your time and efforts in reviewing our paper. Thank you and the reviewers for giving us excellent comments, which helped us significantly improve our manuscript.

Below are our responses to helpful comments and suggestions.

Journal Requirements:

“This study is supported by Ton Duc Thang University. None of the authors receive specific funding numbers for this study.”

Reviewers' comments:

Reviewer's Responses to Questions

Comments to the Author

1. Is the manuscript technically sound, and do the data support the conclusions?

Reviewer #1: Partly

Reviewer #2: Partly

2. Has the statistical analysis been performed appropriately and rigorously?

Reviewer #1: Yes

Reviewer #2: Yes

3. Have the authors made all data underlying the findings in their manuscript fully available?

Reviewer #1: No

Reviewer #2: Yes

4. Is the manuscript presented in an intelligible fashion and written in standard English?

Reviewer #1: Yes

Reviewer #2: Yes

5. Review Comments to the Author

Reviewer #1: The authors provide a worthy contribution to the dense discussion on the impact of cash holdings and financial constraints on firms' bankruptcy risks. Specifically, they focus on 25,002 firm-year observations in the period of 2010-2023, examining only publicly listed A-share companies on the Shanghai and Shenzen stock exchanges. The findings they present align with the expectations of both the market timing theory and the trade-off theory - large financially constrained firms with large cash holdings exhibit lower bankruptcy risk with their Z-score (the primary measure of risk in the paper) rising by ~2.2 points (going by the GMM estimations). The paper is well-structured, concise and tries to lend scrutiny to the authors' findings. However, certain parts of the paper could use improvement.

1. Zhang et al. [9] provides an analysis similar to yours, but only for the Shanghai stock exchange. Your results corroborate their findings, however I would also be interested if there are any differences in the effects you find across the Shanghai and Shenzen stock exchanges. Is the "listing location" something that is available to you? Ideally, you can include a firm's location proxy in the regressions as an additional fixed effect, or you can create heterogeneous effects based on these.

Our responses: We sincerely appreciate the insightful suggestions. We have examined subsamples from the Shanghai and Shenzhen stock exchanges; however, our findings do not indicate statistically significant differences in the effects. The results are presented in Table 8 below.

2. As a robustness measure you introduce the adjusted AWW index along with the KZ index which are both proxies for financial constraint. However, you omit the AWW from table 2 and table 6. What is the reasoning behind this?

Our responses: We thank anonymous reviewers for their valuable feedback. Initially, we considered AKZ and AWW indexes in our primary analysis but decided to focus on AKZ as the primary proxy while using AWW in the robustness test. Since different proxies capture varying aspects of financial constraints, incorporating AWW in the robustness test helps verify our findings' consistency and strengthens our conclusions' reliability. The revised section is Pages 18-20, highlighted in the manuscript.

3. You note: "We also follow Le et al. [24] to winsorize all variables at the 5% and 95% levels to minimize extreme values bias". I consider a winsor that applies to 10% of the observations you use to be at the very least unrepresentative by itself, regardless of the paper you refer to, or the amount of the data at your disposal. Winsoring the data is of course permitted, but I would advise against going over the 2.5% border on either side. Also, you maintain a great deal of the observations you start with even with the usual lag observation losses in GMM - this allows you to trim your sample and avoid any potential confusion. I would wish to see:

• the regression results without any winsoring/trimming,

• the results when winsoring the data on the top and bottom 1% symmetrically, and

• the results when trimming the observations the same way.

Ideally, one of these should be the main paper estimates, or you should provide a more sufficient reason why you winsor on a total of 10% of observations.

Our response: We appreciate the valuable suggestion. To address this concern, we have conducted regressions without winsorizing or trimming and have chosen these results as our main estimates. This approach ensures that our findings are not influenced by data adjustments while maintaining the robustness of our analysis.

4. You split the firms into the manufacturing and non-manufacturing industries, you use the ISG (industry sales growth) in your financial constraint index creation, and you use the industry fixed effects in the regressions. However, you do not note what industry variable you base this on? Is this always a 2-digit sector, as in Chan et al. [5] you reference, or is this something else? This should be made clear and consistent throughout the paper.

Our response: We thank anonymous reviewers for their valuable feedback. In our previous manuscript, the sector classification used in our analysis is consistent with the two-digit sector classification referenced in Chan et al. [5].

5. Another point regarding your split on the manufacturing and non-manufacturing groups of firms. It is true that your results align with the Altman's paper, however your observation count in table 6 is rather odd. 18,367 vs. 1,729 observations seems a quite unbalanced comparison. How do you actually split the firms into these two groups, and is there any way you could improve this split (in terms of observation count balance) with the data available to you?

Our response: We thank anonymous reviewers for their valuable feedback. Following the reviewer's feedback, we realized that the difference between the two industry-based subsamples was too substantial. Therefore, we decided not to divide the firms by industry in the revised analysis. Instead, we follow Fama and French (1993) to split the data into two subsamples based on firm size. The revised section is in Table 7, highlighted in the manuscript.

6. My last major point is your conclusion. Throughout your paper you note that (based on the literature so far) SME's and large firms differ in the effects financial constraints and cash holdings have on the bankruptcy risk of said firms. I assume, since you have not provided any non-ratio descriptive statistics on the firms in your sample, that A-share listed companies are by and large non-SME's. However, you conclude with the practical implications that pertain exactly to SMEs. I would be wary of such interpretations without any "raw" descriptive statistics nor SME-specific heterogeneous effects on the firms in your sample. You should provide the readers with more context on the firms you work with, at the very least for all the starting variables you use in your analysis before any ratios.

Our response: We sincerely appreciate valuable and insightful comments that have helped us improve the quality of our manuscript. Following Fama and French (1993), we classify firms by size: small (below 30th percentile), medium (30th–40th), and large (above 40th). We then group small and medium firms as SMEs to analyze their distinct effects, ensuring our conclusions align with the sample context.

Minor notes:

• You write the 3.4. section as if it was done in advance. In my opinion it makes it only harder to read and understand, with little benefit to it.

Our responses: We thank anonymous reviewers for their valuable feedback. We have revised Section 3.4 to improve clarity and ensure the methodology is more structured and readable. The updated section is on Page 13, highlighted in the manuscript.

• You have two H2 hypotheses, I don't know if this is intentional or not, but I suggest you change it to H2 and H3.

Our responses: We thank anonymous reviewers for their valuable feedback. This issue was unintentional, and I have corrected it by renaming the second H2 to H3 for clarity and consistency. The revised hypotheses are on Page 8 and Page 10, highlighted in the manuscript.

• "We select the sampling period is from 2010 to prevents tha adverse impacts of the financial crisis in 2008 on the findings." - typo on pg. 3 and pg. 8.

Our responses: We thank an anonymous reviewer for identifying this typo. We have corrected it to ensure clarity and accuracy in the text. The updated sentences are on Pages 3 and 11, highlighted in the manuscript.

Reviewer #2: The introduction section is poorly managed. It is likely a literature review. It requires rewriting: The introduction section of a typical research is, needless to say, like a driving force that leads to the rest of the paper and thus it should be carefully written. The introduction should properly present (in no particular order but in a logical manner) the background information, the motivation of the study, the research questions, the gaps in the literature that the study fills, contribution of the study and so on. It is suggested to rewrite the introduction in order to make it sufficient, better sequenced and having read smoothly for the readers.

Our responses: Thank you for your detailed feedback. We have restructured and rewritten the introduction to ensure a more precise flow. The revised section is on Pages 2-5, highlighted in the manuscript.

The authors have discussed two alternative theories supporting the relationship between cash holdings and bankruptcy risk (hypothesis 1). However, they propose a single relationship in the hypothesis. I think that it is better to propose two alternative hypotheses relating to each theory. The same case for hypothesis 2.

Our responses: We thank anonymous reviewers for their valuable feedback. We have revised the hypotheses to reflect the alternative theories by splitting them into H1A, H1B, H2A, and H2B for better clarity and alignment with the theoretical framework. The revised hypotheses are on Page 8 and Page 10, highlighted in the manuscript.

The author(s) need to link their findings more strongly to context, highlight their economic, academic/research, and policy/practice implications.

Our responses: We thank anonymous reviewers for their valuable feedback. We have revi

---

## [Decision Letter · Decision Letter 1]

23 Jun 2025

Dear Dr. Ho,

Thank you for submitting your manuscript to PLOS ONE. After careful consideration, we feel that it has merit but does not fully meet PLOS ONE’s publication criteria as it currently stands. Therefore, we invite you to submit a revised version of the manuscript that addresses the points raised during the review process.

We look forward to receiving your revised manuscript.

Kind regards,

Dariusz Siudak, Ph.D., DSc.

Academic Editor

PLOS ONE

Journal Requirements:

Reviewers' comments:

Reviewer's Responses to Questions

**Comments to the Author**

Reviewer #1: (No Response)

Reviewer #3: (No Response)

2. Is the manuscript technically sound, and do the data support the conclusions?

Reviewer #1: Partly

Reviewer #3: Yes

3. Has the statistical analysis been performed appropriately and rigorously?

Reviewer #1: Yes

Reviewer #3: Yes

4. Have the authors made all data underlying the findings in their manuscript fully available?

Reviewer #1: No

Reviewer #3: Yes

5. Is the manuscript presented in an intelligible fashion and written in standard English?

Reviewer #1: Yes

Reviewer #3: Yes

Reviewer #1: The authors have, in my opinion, made reasonably sufficient improvements in response to my previous comments. However, I still have a few hanging questions. My comments here follow the same order (numbers) as the previous review.

1.

Thank you for addressing my suggestion on the location split and for conducting the subsample analysis. Given that the results do not indicate statistically significant differences, I recommend moving them to the appendix rather than including them in the main findings.

2.

I respect your choice not to include the AWW in your main results. However, since AWW is still used to validate the robustness of your findings, I recommend including it in the Table 2 correlation matrix to provide a more comprehensive overview of the relationships among the key variables, as AWW is one too.

5.

Thank you for reconsidering the industry-based split and for adopting the Fama and French (1993) approach based on firm size. This is clearly a more suitable alternative in your case. However, I would advise against labeling these firms as SMEs, as the SME classification is often based on the EU definition, which differs from this approach. This clarification would help avoid potential misinterpretations.

6.

Again, incorporating the size-based classification following Fama and French (1993) to refine your analysis is more in line with your findings and your research framework in general. However, while this approach is more suitable, it still relies on a ratio-based sample split. Beyond the age of the firm, you have not provided any non-ratio descriptive statistics for the firms in your sample. Is there a specific reason for this omission? Providing such descriptives would offer readers more context on the firms analyzed and strengthen the validity of your conclusions, particularly regarding SMEs.

Reviewer #3: The paper provides important insights about the association between financial constraints and corporate bankruptcy risks. However, some areas still need improvements.

- For the third hypothesis, author(s) should elaborate more on the interactive effect of financial constraints and cash holding on bankruptcy risk as this issue represents a focal point in the current research.

- In table 1 descriptive statistics it is noted that the mean of leverage ratio (the ratio of total debt to total assets) is 243.2 % which is out of logic. The same also for the median, max and min values. Please check and confirm the descriptive statistics for leverage ratio.

- The results of Table 8 that presents the robustness findings are not discussed in the “Discussion section” in the main text. Please refer to the results of table 8 in the discussion section.

- In page 23 author(s) indicate that the results of table 5 show that leverage positively impact the probability of default risk. However, the results in table 5 do not show positive impact, but rather indicate that increases in leverage decrease distress risk.

**Do you want your identity to be public for this peer review?** For information about this choice, including consent withdrawal, please see our Privacy Policy

Reviewer #1: No

Reviewer #3: No

---

## [Author Response · Author response to Decision Letter 2]

4 Aug 2025

PONE-D-24-39978R1

FINANCIAL CONSTRAINTS AND CORPORATE BANKRUPTCY RISKS IN CHINA: THE BUFFER ROLE OF CASH HOLDINGS

PLOS ONE

Journal Requirements:

Comments to the Author

1. If the authors have adequately addressed your comments raised in a previous round of review and you feel that this manuscript is now acceptable for publication, you may indicate that here to bypass the “Comments to the Author” section, enter your conflict of interest statement in the “Confidential to Editor” section, and submit your "Accept" recommendation.

Reviewer #1: (No Response)

Reviewer #3: (No Response)

2. Is the manuscript technically sound, and do the data support the conclusions?

Reviewer #1: Partly

Reviewer #3: Yes

3. Has the statistical analysis been performed appropriately and rigorously?

Reviewer #1: Yes

Reviewer #3: Yes

4. Have the authors made all data underlying the findings in their manuscript fully available?

Reviewer #1: No

Reviewer #3: Yes

5. Is the manuscript presented in an intelligible fashion and written in standard English?

Reviewer #1: Yes

Reviewer #3: Yes

6. Review Comments to the Author

Reviewer #1: The authors have, in my opinion, made reasonably sufficient improvements in response to my previous comments. However, I still have a few hanging questions. My comments here follow the same order (numbers) as the previous review.

1. Thank you for addressing my suggestion on the location split and for conducting the subsample analysis. Given that the results do not indicate statistically significant differences, I recommend moving them to the appendix rather than including them in the main findings.

Our responses: We thank anonymous reviewers for their valuable feedback. Following the reviewer's feedback, we have moved Table 8 to Appendix B.

2. I respect your choice not to include the AWW in your main results. However, since AWW is still used to validate the robustness of your findings, I recommend including it in the Table 2 correlation matrix to provide a more comprehensive overview of the relationships among the key variables, as AWW is one too.

Our responses: We sincerely appreciate the insightful suggestions. Following the reviewer's feedback, we have added AWW into the Table 2 correlation matrix. The revised section is in Table 2, highlighted in the manuscript.

5. Thank you for reconsidering the industry-based split and for adopting the Fama and French (1993) approach based on firm size. This is clearly a more suitable alternative in your case. However, I would advise against labeling these firms as SMEs, as the SME classification is often based on the EU definition, which differs from this approach. This clarification would help avoid potential misinterpretations.

Our responses: We sincerely appreciate the insightful suggestions. Following the reviewer's feedback, we have changed "SMEs" to "Small and Medium-sized firms" to avoid potential misinterpretations.

6. Again, incorporating the size-based classification following Fama and French (1993) to refine your analysis is more in line with your findings and your research framework in general. However, while this approach is more suitable, it still relies on a ratio-based sample split. Beyond the age of the firm, you have not provided any non-ratio descriptive statistics for the firms in your sample. Is there a specific reason for this omission? Providing such descriptives would offer readers more context on the firms analyzed and strengthen the validity of your conclusions, particularly regarding SMEs.

Our responses: Thank you for your valuable comment. We focused on ratio-based variables to ensure unit consistency with our dependent variable (Z-score), which is a standardized measure. Including non-ratio descriptives could complicate interpretation due to scale differences.

Reviewer #3: The paper provides important insights about the association between financial constraints and corporate bankruptcy risks. However, some areas still need improvements.

- For the third hypothesis, author(s) should elaborate more on the interactive effect of financial constraints and cash holding on bankruptcy risk as this issue represents a focal point in the current research.

Our responses: Thank you for your suggestion. We have revised the section to elaborate on how cash holdings can moderate the impact of financial constraints on bankruptcy risk. The revised section is highlighted on pages 10 and 11.

- In table 1 descriptive statistics it is noted that the mean of leverage ratio (the ratio of total debt to total assets) is 243.2 % which is out of logic. The same also for the median, max and min values. Please check and confirm the descriptive statistics for leverage ratio.

Our responses: Thank you for pointing out the issue with the leverage ratio. Upon review, we identified an error in the calculation of this variable. As a result, we have removed leverage from our analysis and updated the manuscript accordingly. We appreciate your careful attention to detail.

- The results of Table 8 that presents the robustness findings are not discussed in the “Discussion section” in the main text. Please refer to the results of table 8 in the discussion section.

Our responses: Thank you for the comment. We have now referred to the results of Table 8 in the Discussion section and briefly explained the consistency across both exchanges. As suggested, Table 8 has been moved to Appendix B.

- In page 23 author(s) indicate that the results of table 5 show that leverage positively impact the probability of default risk. However, the results in table 5 do not show positive impact, but rather indicate that increases in leverage decrease distress risk.

Our responses: Thank you for your careful reading. Upon review, we found an error in the interpretation and the calculation of the leverage variable. Due to this issue, we have excluded leverage variable from our final model and updated the manuscript accordingly.

---

## [Decision Letter · Decision Letter 2]

27 Aug 2025

Dear Dr. Ho,

We look forward to receiving your revised manuscript.

Kind regards,

Clinton Watkins, Ph.D.

Academic Editor

PLOS ONE

Journal Requirements:

Additional Editor Comments:

Please respond to the remaining comment by Reviewer 3 to the manuscript Revision 2.

Reviewers' comments:

Reviewer's Responses to Questions

**Comments to the Author**

Reviewer #1: (No Response)

Reviewer #3: (No Response)

2. Is the manuscript technically sound, and do the data support the conclusions?

Reviewer #1: Yes

Reviewer #3: Yes

3. Has the statistical analysis been performed appropriately and rigorously?

Reviewer #1: Yes

Reviewer #3: Yes

4. Have the authors made all data underlying the findings in their manuscript fully available?

Reviewer #1: Yes

Reviewer #3: Yes

5. Is the manuscript presented in an intelligible fashion and written in standard English?

Reviewer #1: Yes

Reviewer #3: Yes

Reviewer #1: The authors have addressed my previous comments with reasonable improvements. Below, I provide my final remarks, following my earlier points.

1. Thank you for examining the differences between the exchanges. If you prefer, you may place table 8 in the appendix, as no substantial conclusions can be drawn from it.

2. I appreciate the authors’ rationale regarding the approach to the main results. While I would still prefer to see AWW included in table 2, I acknowledge that this is not critical to the decision on the paper’s acceptance, so I will not insist on it.

3. Thank you for omitting the winsorization. This approach is more transparent. If you wish to do so, the original results could be given in an online appendix for reference.

4. Thank you for the clarification provided.

5. The revised split appears to be more accurate, and I appreciate the decision to move away from the previous version.

6. The conclusion now carries greater validity, as the SME classification is better justified. Nevertheless, the classification of SME's is still not the most widely used one, and raw descriptive statistics are not provided. I will not insist on these, but both the authors and editors should be aware of this limitation.

Reviewer #3: Thank you for considering the comments. However, regarding the calculations for leverage ratio (comment 3), authors indicate that they identified an error in the calculation of this variable (leverage ratio). As a result, they have removed leverage from analysis and updated the manuscript accordingly. However, the variable Leverage is still included into the calculations of variables (Z_FC and AWW). Please confirm that the leverage ratio used in the calculations of the mentioned variables is right.

**Do you want your identity to be public for this peer review?** For information about this choice, including consent withdrawal, please see our Privacy Policy

Reviewer #1: No

Reviewer #3: No

---

## [Author Response · Author response to Decision Letter 3]

9 Sep 2025

PONE-D-24-39978R2

FINANCIAL CONSTRAINTS AND CORPORATE BANKRUPTCY RISKS IN CHINA: THE BUFFER ROLE OF CASH HOLDINGS

Review Comments to the Author

Reviewer #1: The authors have addressed my previous comments with reasonable improvements. Below, I provide my final remarks, following my earlier points.

1. Thank you for examining the differences between the exchanges. If you prefer, you may place Table 8 in the appendix, as no substantial conclusions can be drawn from it.

Our response: We sincerely appreciate the insightful suggestions. Following the reviewer's feedback, we have moved Table 8 to Appendix B in the previous manuscript.

2. I appreciate the authors’ rationale regarding the approach to the main results. While I would still prefer to see AWW included in Table 2, I acknowledge that this is not critical to the decision on the paper’s acceptance, so I will not insist on it.

Our response: Thank you for your valuable comment. We have decided to keep AWW included in Table 2.

3. Thank you for omitting the winsorization. This approach is more transparent. If you wish to do so, the original results could be given in an online appendix for reference.

Our response: Thank you for your thoughtful suggestion. At this stage, we prefer not to include the original results in an online appendix, as we believe the revised version better reflects the focus of our study.

4. Thank you for the clarification provided.

Our response: We sincerely thank anonymous reviewers.

5. The revised split appears to be more accurate, and I appreciate the decision to move away from the previous version.

Our response: We sincerely thank anonymous reviewers.

6. The conclusion now carries greater validity, as the SME classification is better justified. Nevertheless, the classification of SME's is still not the most widely used one, and raw descriptive statistics are not provided. I will not insist on these, but both the authors and editors should be aware of this limitation.

Our response: We sincerely thank the anonymous reviewer for this constructive comment. We acknowledge that the SME classification based on the Fama and French (1993) size breakpoint is not the most widely used approach, particularly in the context of China. In response, we have explicitly acknowledged this limitation in the conclusion. The revised section is highlighted on page 27.

Reviewer #3: Thank you for considering the comments. However, regarding the calculations for leverage ratio (comment 3), authors indicate that they identified an error in the calculation of this variable (leverage ratio). As a result, they have removed leverage from analysis and updated the manuscript accordingly. However, the variable Leverage is still included into the calculations of variables (Z_FC and AWW). Please confirm that the leverage ratio used in the calculations of the mentioned variables is right.

Our response: Thank you for your valuable comment. We confirm that the leverage ratio used in the calculations of AWW and Z_FC proxies has been correctly computed.

---

## [Decision Letter · Decision Letter 3]

12 Oct 2025

Dear Dr. Ho,

Thank you for submitting your manuscript to PLOS ONE. After careful consideration, we feel that it has merit but does not fully meet PLOS ONE’s publication criteria as it currently stands. Therefore, we invite you to submit a revised version of the manuscript that addresses the points raised during the review process.

While both reviewers recommended acceptance, I have carefully examined the paper as an editor and identified several major concerns that must be addressed before the manuscript can be considered further. These revisions are essential to improve the theoretical rigor, methodological transparency, and overall scholarly contribution of the study.

Below, I outline the key points that require your attention. Please respond comprehensively to each issue in your revision and accompanying response letter.

1. The manuscript’s theoretical grounding is superficial and fragmented. The discussion of pecking order, trade-off, agency, and market-timing theories is descriptive rather than integrative. The link between these theories and the hypothesized relationships is weak, with no clear conceptual mechanism explaining how cash holdings buffer financial constraints.

Develop a unified conceptual framework showing the causal pathways linking financial constraints, cash holdings, and bankruptcy risk. Explicitly justify the buffer mechanism theoretically—e.g., through liquidity risk mitigation, precautionary saving behavior, or financing flexibility channels.

2. Presenting each hypothesis with both positive and negative expected directions (e.g., H1A/H1B, H2A/H2B) is conceptually meaningless. It indicates the absence of a clear theoretical stance and weakens the research design.

Reformulate each hypothesis with a single, theory-driven directional expectation grounded in the literature. Remove the dual-sided hypothesis structure entirely.

3. The literature review is overly descriptive and outdated, relying heavily on older studies and lacking synthesis of recent (post-2020) works on liquidity buffers, macroprudential regulation, and bankruptcy prediction in China.

Condense repetitive sections, integrate recent literature (2021–2024), and structure the review thematically (theory → empirical contradictions → research gap). Emphasize what distinguishes the Chinese institutional setting and how this context strengthens your contribution.

4. The Z_FC and AWW indices are used without verifying their suitability for the Chinese context. The manuscript also claims to have removed Leverage due to a calculation error but still retains it in both indices, causing inconsistency.

Validate these indices empirically (e.g., via correlation or PCA tests) to confirm their relevance for Chinese firms. Ensure internal consistency by either recalculating the indices without Leverage or clearly justifying its inclusion.

5. The justification for employing dynamic system GMM is inadequate. The inclusion of the lagged Z-score is not theoretically motivated, and the use of 78 instruments risks overfitting. Additionally, the manuscript reports the J-statistic instead of the Hansen test, which is more robust for over-identification.

Explain why the dynamic specification is required (e.g., persistence in financial distress). Limit the number of instruments (ideally fewer than the number of groups) to avoid weak identification. Replace the J-statistic with the Hansen test and report AR(1) and AR(2) diagnostics explicitly. Discuss instrument validity in the text.

6. The fixed effects are inconsistently applied across models. FEM includes industry and year dummies, but the GMM section does not specify whether these are retained.

Apply a consistent fixed-effects structure across all models (industry and year) or clearly justify any deviation.

7. The manuscript identifies endogenous variables but does not explain the instrument selection strategy or lag structure. Robustness checks are limited to alternative proxies and firm-size subsamples, omitting alternative distress measures.

Describe the choice and lag depth of instruments in detail. Add at least one alternative bankruptcy risk measure (e.g., O-score, Altman 1993 revision, or distance-to-default) to enhance robustness.

8. The discussion primarily repeats coefficient signs without exploring their economic meaning. The interaction term (Z_FC × CASH) lacks visual or quantitative illustration.

Include marginal-effect or interaction plots to demonstrate moderation effects. Discuss economic significance and managerial relevance rather than only statistical results.

9. Heteroskedasticity and serial correlation are noted but not addressed in sufficient detail.

Discuss the implications of these diagnostic results. If necessary, rerun GMM with collapsed instruments or shorter lags to improve test performance. Specify whether standard errors are robust or clustered and justify this choice.

10. The paper contains redundant sentences, grammatical issues, and inconsistent variable names.

Undertake a professional English edit, eliminate redundancy, and ensure consistent names. Revise the abstract to emphasize motivation, methods, main results, and contribution succinctly.

11. The conclusion reiterates findings without reflecting on their implications. Theoretical and policy contributions are generic, and limitations are not tied to future research directions.

Reframe the conclusion to emphasize (a) theoretical advancement in understanding liquidity–distress dynamics, (b) managerial and policy relevance for emerging markets, and (c) clear directions for future comparative or institutional research.

We look forward to receiving your revised manuscript.

Kind regards,

Islam Abdeljawad

Academic Editor

PLOS ONE

Journal Requirements:

Reviewers' comments:

Reviewer's Responses to Questions

**Comments to the Author**

Reviewer #1: All comments have been addressed

Reviewer #3: All comments have been addressed

2. Is the manuscript technically sound, and do the data support the conclusions?

Reviewer #1: Yes

Reviewer #3: Yes

3. Has the statistical analysis been performed appropriately and rigorously?

Reviewer #1: Yes

Reviewer #3: Yes

4. Have the authors made all data underlying the findings in their manuscript fully available?

Reviewer #1: Yes

Reviewer #3: Yes

5. Is the manuscript presented in an intelligible fashion and written in standard English?

Reviewer #1: Yes

Reviewer #3: Yes

Reviewer #1: The authors have, in my opinion, made sufficient improvements in response to all previous comments.

Reviewer #3: (No Response)

**Do you want your identity to be public for this peer review?** For information about this choice, including consent withdrawal, please see our Privacy Policy

Reviewer #1: No

Reviewer #3: No

---

## [Author Response · Author response to Decision Letter 4]

27 Dec 2025

PONE-D-24-39978R3

FINANCIAL CONSTRAINTS AND CORPORATE BANKRUPTCY RISKS IN CHINA: THE BUFFER ROLE OF CASH HOLDINGS

While both reviewers recommended acceptance, I have carefully examined the paper as an editor and identified several major concerns that must be addressed before the manuscript can be considered further. These revisions are essential to improve the theoretical rigor, methodological transparency, and overall scholarly contribution of the study.

Below, I outline the key points that require your attention. Please respond comprehensively to each issue in your revision and accompanying response letter.

1. The manuscript’s theoretical grounding is superficial and fragmented. The discussion of pecking order, trade-off, agency, and market-timing theories is descriptive rather than integrative. The link between these theories and the hypothesized relationships is weak, with no clear conceptual mechanism explaining how cash holdings buffer financial constraints.

Develop a unified conceptual framework showing the causal pathways linking financial constraints, cash holdings, and bankruptcy risk. Explicitly justify the buffer mechanism theoretically—e.g., through liquidity risk mitigation, precautionary saving behavior, or financing flexibility channels.

Our response: We have significantly revised the Literature Review to move beyond a descriptive list of theories. We now present a unified conceptual framework that integrates Market Timing, Trade-off, and Precautionary Savings theories. The revised Literature Review section is highlighted on pages 5-12.

2. Presenting each hypothesis with both positive and negative expected directions (e.g., H1A/H1B, H2A/H2B) is conceptually meaningless. It indicates the absence of a clear theoretical stance and weakens the research design.

Reformulate each hypothesis with a single, theory-driven directional expectation grounded in the literature. Remove the dual-sided hypothesis structure entirely.

Our response: We thank the reviewer for the critique regarding our hypothesis structure. We have completely removed the dual-sided structure (H1A/H1B) and reformulated our hypotheses to reflect a single, theory-driven stance. The revised Literature Review section is highlighted on pages 5-12.

3. The literature review is overly descriptive and outdated, relying heavily on older studies and lacking synthesis of recent (post-2020) works on liquidity buffers, macroprudential regulation, and bankruptcy prediction in China.

Condense repetitive sections, integrate recent literature (2021–2024), and structure the review thematically (theory → empirical contradictions → research gap). Emphasize what distinguishes the Chinese institutional setting and how this context strengthens your contribution.

Our response: We appreciate the reviewer’s guidance on modernizing our literature review. We have significantly condensed repetitive theoretical descriptions and restructured the review thematically with new references. The revised Literature Review section is highlighted on pages 5-12.

4. The Z_FC and AWW indices are used without verifying their suitability for the Chinese context. The manuscript also claims to have removed Leverage due to a calculation error but still retains it in both indices, causing inconsistency.

Validate these indices empirically (e.g., via correlation or PCA tests) to confirm their relevance for Chinese firms. Ensure internal consistency by either recalculating the indices without Leverage or clearly justifying its inclusion.

Our response: We appreciate the editor’s comments regarding the suitability of the Z_FC and AWW indices and the inconsistency involving Leverage. To address these fundamental concerns, we have entirely replaced the previous indices with the SA Index (Hadlock & Pierce, 2010) as the sole proxy for financial constraints in the revised manuscript. We selected the SA Index because it relies exclusively on firm size and age, which are relatively exogenous variables, thereby avoiding the endogeneity problems inherent in leverage-based measures like the Z_FC or AWW. The revised section is highlighted on page 13.

Consequently, this substitution resolves the inconsistency issue, as Leverage is no longer included in the constraint construction nor the regression model.

5. The justification for employing dynamic system GMM is inadequate. The inclusion of the lagged Z-score is not theoretically motivated, and the use of 78 instruments risks overfitting. Additionally, the manuscript reports the J-statistic instead of the Hansen test, which is more robust for over-identification.

Explain why the dynamic specification is required (e.g., persistence in financial distress). Limit the number of instruments (ideally fewer than the number of groups) to avoid weak identification. Replace the J-statistic with the Hansen test and report AR(1) and AR(2) diagnostics explicitly. Discuss instrument validity in the text.

Our response: We appreciate the editor’s valuable feedback on the econometric specification. In the revised manuscript, we have strengthened the justification for the GMM estimation, arguing that bankruptcy risk exhibits persistence; specifically, past financial distress heavily conditions current risk due to reputation effects and long-term debt obligations, making the inclusion of the lagged dependent variable theoretically essential.

To address the concern of overfitting, we implemented the collapse option and restricted lag depths, ensuring the instrument count (45) remains strictly below the number of firms (groups), thereby improving the power of the overidentification test.

Furthermore, we have replaced the generic J-statistic with the Hansen test, and we explicitly report the AR(1) and AR(2) diagnostics. The insignificant Hansen and AR(2) results confirm the validity of our instruments and the absence of second-order serial correlation.

6. The fixed effects are inconsistently applied across models. FEM includes industry and year dummies, but the GMM section does not specify whether these are retained.

Apply a consistent fixed-effects structure across all models (industry and year) or clearly justify any deviation.

Our response: We sincerely thank the editor for pointing out the inconsistency in the fixed effects structure and acknowledge this oversight in the previous version. In the revised manuscript, we have excluded year and industry fixed effects from the fixed-effects estimation in the revised manuscript to align with the specific econometric properties of each estimator. However, for the Two-step system GMM estimation, we use both industry and year fixed effects (i.industry and i.year) in the instrument matrix. This deviation is justified because the Two-step system GMM allows us to control for sector-specific heterogeneity and temporal macroeconomic shocks rigorously.

7. The manuscript identifies endogenous variables but does not explain the instrument selection strategy or lag structure. Robustness checks are limited to alternative proxies and firm-size subsamples, omitting alternative distress measures.

Describe the choice and lag depth of instruments in detail. Add at least one alternative bankruptcy risk measure (e.g., O-score, Altman 1993 revision, or distance-to-default) to enhance robustness.

Our response: We sincerely thank the editor for this meticulous and constructive comment. We have substantially revised the methodology to detail our instrument selection strategy transparently, specifying that we treat the lagged dependent variable and key financial regressors as endogenous, instrumented using their own specific lag depths (strictly restricted to lags 2-2, and lags 3-3 for the interaction term) to address simultaneity while using the 'collapse' option to prevent instrument proliferation. Strictly exogenous variables (e.g., FAT, industry, and year effects) are included in the standard IV matrix. Furthermore, to address the concern regarding limited robustness, we have expanded our analysis by re-estimating the model using three alternative bankruptcy risk measures: the ZM-score, Z'-score, and Z"-score; the consistency of these results with our baseline findings significantly strengthens the validity of our conclusions regarding liquidity–distress dynamics.

8. The discussion primarily repeats coefficient signs without exploring their economic meaning. The interaction term (Z_FC × CASH) lacks visual or quantitative illustration.

Include marginal-effect or interaction plots to demonstrate moderation effects. Discuss economic significance and managerial relevance rather than only statistical results.

Our response: We appreciate the editor’s constructive suggestion to deepen the interpretation of the interaction results. Following your feedback, we have introduced Table 6 and a marginal-effect plot (Figure 1) that visually demonstrates how cash holdings moderate the impact of financial constraints on bankruptcy risk across different liquidity zones. Furthermore, we have thoroughly revised the discussion to prioritize economic significance and managerial relevance, explicitly interpreting the reversal in marginal effects as evidence of cash serving as a strategic stability buffer and a decoupling mechanism, rather than merely reporting statistical coefficient signs. The revised section is highlighted on pages 19-20.

9. Heteroskedasticity and serial correlation are noted but not addressed in sufficient detail.

Discuss the implications of these diagnostic results. If necessary, rerun GMM with collapsed instruments or shorter lags to improve test performance. Specify whether standard errors are robust or clustered and justify this choice.

Our response: We thank the editor for the technical scrutiny regarding the GMM specification. We confirm that our estimation strategy explicitly addresses these diagnostic concerns through the specific options employed in the xtabond2 command.

As detailed in our revised methodology, we employ the Two-step System GMM estimator with the robust option. In the context of xtabond2, this automatically reports standard errors that are robust to heteroskedasticity and clustered at the firm level. Crucially, this option applies the Windmeijer (2005) finite-sample correction to the standard errors, ensuring that the test statistics are reliable and not biased downwards, which is a common issue in two-step estimation.

Furthermore, we utilized the collapse option and strictly limited lag depths (specifically lag(2 2) for main variables and lag(3 3) for the interaction term) to prevent instrument proliferation and preserve the power of the Hansen test.

We report the AR(1) and AR(2) test results in the "Empirical Results" section, interpreting the significant AR(1) and insignificant AR(2) as evidence validating the absence of serial correlation in the error terms.

10. The paper contains redundant sentences, grammatical issues, and inconsistent variable names.

Undertake a professional English edit, eliminate redundancy, and ensure consistent names. Revise the abstract to emphasize motivation, methods, main results, and contribution succinctly.

Our response: We sincerely appreciate the feedback on presentation quality and have undertaken professional proofreading to correct errors, remove redundancy, and ensure variable consistency throughout the manuscript. Additionally, we have completely revised the Abstract to succinctly highlight the study's motivation, methodology, main results, and contributions as requested.

11. The conclusion reiterates findings without reflecting on their implications. Theoretical and policy contributions are generic, and limitations are not tied to future research directions.

Reframe the conclusion to emphasize (a) theoretical advancement in understanding liquidity–distress dynamics, (b) managerial and policy relevance for emerging markets, and (c) clear directions for future comparative or institutional research.

Our responses: We sincerely appreciate the editor’s insightful and constructive feedback regarding the Conclusion section. We acknowledge that the previous version was primarily descriptive and lacked a deep discussion of implications. Following your guidance, we have completely rewritten the Conclusion to explicitly highlight the theoretical advancements in liquidity–distress dynamics, discuss specific managerial and policy implications for emerging markets like China, and link our limitations to meaningful avenues for future institutional research. The revised section is highlighted on pages 29-32.

References

• Hadlock CJ, Pierce JR. New Evidence on Measuring Financial Constraints: Moving Beyond the KZ Index. Rev Financ Stud. 2010;23: 1909–1940. doi:10.1093/rfs/hhq009

• Windmeijer F. A finite sample correction for the variance of linear efficient two-step GMM estimators. Journal of Econometrics. 2005;126: 25–51. doi:10.1016/j.jeconom.2004.02.005

---

## [Editor Report · Decision Letter 4]

4 Jan 2026

FINANCIAL CONSTRAINTS AND CORPORATE BANKRUPTCY RISKS IN CHINA: THE BUFFER ROLE OF CASH HOLDINGS

PONE-D-24-39978R4

Dear Dr. Ho,

We’re pleased to inform you that your manuscript has been judged scientifically suitable for publication and will be formally accepted for publication once it meets all outstanding technical requirements.

Kind regards,

Islam Abdeljawad

Academic Editor

PLOS One

Additional Editor Comments (optional):

All major concerns have been addressed, and the manuscript is acceptable.
---

## [Editor Report · Acceptance letter]

PONE-D-24-39978R4

PLOS One

Dear Dr. Ho,

I'm pleased to inform you that your manuscript has been deemed suitable for publication in PLOS One. Congratulations! Your manuscript is now being handed over to our production team.

Kind regards,

on behalf of

Dr. Islam Abdeljawad

Academic Editor

PLOS One